# Integrating when and what information in the left parietal lobe allows language rule generalization

Joan Orpella[1,2,3,4☯], Pablo Ripollés[4,5,6☯], Manuela Ruzzoli[7,8], Julià L. Amengual[9], Alicia Callejas[1,10], Anna Martinez-Alvarez[1,2,3,11], Salvador Soto-Faraco[5,12], Ruth de Diego-Balaguer[1,2,3,12]*

**1** Cognition and Brain Plasticity Unit, IDIBELL, L'Hospitalet de Llobregat, Spain, **2** Dept of Cognition Development and Educational Psychology, University of Barcelona, Barcelona, Spain, **3** Institute of Neuroscience, University of Barcelona, Barcelona, Spain, **4** Department of Psychology, New York University, New York, New York, United States of America, **5** Music and Auditory Research Laboratory (MARL), New York University, New York, New York, United States of America, **6** Center for Language, Music and Emotion (CLaME), New York University, New York, New York, United States of America, **7** Center for Brain and Cognition, Departament de Tecnologies de la Informació i les Comunicacions, Universitat Pompeu Fabra, Barcelona, Spain, **8** Institute of Neuroscience and Psychology, University of Glasgow, Glasgow, United Kingdom, **9** Centre de Neuroscience Cognitive Marc Jeannerod, CNRS UMR 5229, Université Claude Bernard Lyon I, Bron, France, **10** Departamento de Psicología Experimental, Facultad de Psicología y Centro de Investigación Mente, Cerebro y Comportamiento, Universidad de Granada, Granada, Spain, **11** Department of Developmental Psychology and Socialization, University of Padua, Italy, **12** ICREA, Barcelona, Spain

☯ These authors contributed equally to this work.
\* ruth.dediego@ub.edu

**Data Availability Statement:** All relevant behavioral data are within the paper and its Supporting Information. Main fMRI contrasts can

## Abstract

A crucial aspect when learning a language is discovering the rules that govern how words are combined in order to convey meanings. Because rules are characterized by sequential co-occurrences between elements (e.g., "**These** cupcake**s are un**believ**able**"), tracking the statistical relationships between these elements is fundamental. However, purely bottom-up statistical learning alone cannot fully account for the ability to create abstract rule representations that can be generalized, a paramount requirement of linguistic rules. Here, we provide evidence that, after the statistical relations between words have been extracted, the engagement of goal-directed attention is key to enable rule generalization. Incidental learning performance during a rule-learning task on an artificial language revealed a progressive shift from statistical learning to goal-directed attention. In addition, and consistent with the recruitment of attention, functional MRI (fMRI) analyses of late learning stages showed left parietal activity within a broad bilateral dorsal frontoparietal network. Critically, repetitive transcranial magnetic stimulation (rTMS) on participants' peak of activation within the left parietal cortex impaired their ability to generalize learned rules to a structurally analogous new language. No stimulation or rTMS on a nonrelevant brain region did not have the same interfering effect on generalization. Performance on an additional attentional task showed that this rTMS on the parietal site hindered participants' ability to integrate "what" (stimulus identity) and "when" (stimulus timing) information about an expected target. The present findings suggest that learning rules from speech is a two-stage process: following statistical

be found at https://identifiers.org/neurovault.collection:8592.

**Funding:** This work was supported by the European Research Council grant ERC-StG-313841 (TuningLang) and the BFU2017-87109-P Grant from the Spanish Ministerio de Ciencia e Innovación (RdD-B) which is part of Agencia Estatal de Investigación (AEI) (Co-funded by the European Regional Development Fund. ERDF, a way to build Europe), the European Research Council (Proof of Concept, ERC, 727595) (SS-F), the Juan de la Cierva Post-Doctorate Fellowship (JCI-2012-12335, Ministerio de Economia y Competividad) (MR) and CERCA Programme / Generalitat de Catalunya for institutional support. The funders had no role in study design, data collection and analysis, decision to publish, or preparation of the manuscript.

**Competing interests:** The authors have declared that no competing interests exist.

**Abbreviations:** BOLD, blood-oxygenation-level–dependent; dlPFC, dorsolateral prefrontal cortex; fMRI, functional MRI; FWE, family-wise error; IFG, inferior frontal gyrus; iPL, inferior parietal lobule; IPS, intraparietal sulcus; ISI, inter-stimulus interval; lPL, left parietal lobe; MNI, Montreal Neurological Institute; POz, midline posterior location according to the 10–20 system electrode location; RT, reaction time; rTMS, repetitive transcranial magnetic stimulation; SPL, superior parietal lobe; TE, echo time; TR, repetition time.

learning, goal-directed attention—involving left parietal regions—integrates "what" and "when" stimulus information to facilitate rapid rule generalization.

## Introduction

Our increasing understanding of the interplay between domain-specific and domain-general cognitive processes has gradually broadened our views on language learning. Apparently simple feats such as the learning of new words are no longer thought to result from the sole workings of a system specialized for language but are known to involve general-purpose mechanisms of statistical learning [1], memory consolidation [2], attention [3], or reward [4]. Besides words, linguistic proficiency requires the learning of and, particularly, the ability to generalize rules, which involves the development of abstract representations of grammatical categories and an understanding of their interrelations. Although this has been the topic of much research, a fundamental question remains unresolved: what are the brain mechanisms that support language rule learning and generalization?

Given its core relevance, the mechanisms supporting rule learning have been subject to intense debate. By their very nature, language rules are characterized by sequential co-occurrences, often between nonadjacent elements (e.g., **These** cupcake**s** **are** **un**believ**able**). Accordingly, the tracking of statistical relations from the input is known to be a key computation in this context [5] as it is in many other domains [6,7]. Yet, purely bottom-up statistical learning alone cannot fully account for the generalization of nonadjacent dependencies [8–10], and an additional role for attention has been posited as necessary [11–17]. The deployment of attention has been shown to be crucial in determining what information will be learned [16] and to bias subsequent learning [18]. Indeed, it is often the case in natural languages that dependencies occur at salient positions such as edges, with edges acting as perceptual anchor points that could additionally help to learn the positional information of the related elements [19]. In that sense, perceptual cues such as pauses make the elements at the edges more salient, which directs attention to these positions facilitating learning. In line with this, attention has been also proposed to underpin the ability to transfer the rules of a first language to a structurally similar second language [20,21], and diverting attention interferes with rule generalization [22]. However, and despite the increasing support for the implication of attention in rule learning and generalization, there is currently no direct evidence for how attention interacts with statistical learning during the extraction of language rules and how this interaction facilitates generalization. Moreover, the brain networks enacting this interaction for language rule learning and generalization remain largely unspecified.

In the current article, we propose that attention is essential in the process leading from the bottom-up extraction of statistical regularities in the input to the abstraction and generalization of rules. In particular, we capitalized on prior research on attention to test the hypothesis that different types of attention interact with statistical learning to support language rule learning and generalization.

Research in the domain of attention posits the distinction between 2 networks for attention orienting [23,24]: a ventral frontoparietal network running along perisylvian areas from the inferior parietal lobule (iPL) to the inferior frontal gyrus (IFG), involved in the automatic detection of behaviorally relevant stimuli [23,24] (that is, stimulus-driven attention), and a more dorsal and bilateral frontoparietal attention network, including the dorsolateral prefrontal cortex (dlPFC) and the intraparietal sulcus (IPS) extending to the superior parietal lobe (SPL) [23], involved in orienting attention towards stimulus features based on internal goals (that is, goal-directed attention).

An important feature of the stimulus-driven ventral attention network is its sensitivity to input statistics, such as the probability of a cue predicting an upcoming target [25]. Critically, this sensitivity is observed both in the spatial and temporal modalities, with the difference that predicting a target at a specific spatial location correlates with right-lateralized activity, whereas predicting when a target will appear elicits a left-lateralized response [26,27]. Given that the relationship between rule elements in speech necessarily involves a temporal dimension, we hypothesized that the bottom-up tracking of statistical relationships for rule learning will predominantly engage left-lateralized stimulus-driven ventral network activity. Supporting this hypothesis, a variety of studies on language rule learning [28–30], as well as on statistical learning for word segmentation from speech [31], report the involvement of a left-lateralized frontoparietal network in the early stages of exposure to a new language, highlighting an overlooked overlap between the left ventral attention network for temporal orienting and the classic language network in its frontoparietal component. Importantly, the ventral network is also thought to inform [24] the dorsal network for the generation and updating of internal models of the environment [25] that will ultimately guide goal-directed attention. Moreover, the right/left specialization in spatial versus temporal domains observed in the ventral network is also observed in the dorsal network [27] for goal-directed attention.

In line with these observations, we hypothesize that rule learning is a two-stage process involving an initial stimulus-driven statistical learning stage, recruiting the (left) ventral attention network, followed by the engagement of goal-directed attention as learning proceeds, associated with the dorsal frontoparietal attention network. Therefore, we expected goal-directed attention to play a more prominent role at later stages of learning and to be critical for rule generalization to new languages by providing the relevant temporal structure. Our specific predictions were that i) the ventral stimulus-driven attention network will be primarily activated in response to rule elements in the learning of their statistical contingency; ii) informed by this, the dorsal attention system will, in turn, direct attention towards these relevant elements of the speech stream, both in terms of their identity (that is, "what" information; e.g., "un" predicting "able" in **un**forgett**able**) and the specific position (that is, moment in time) in which these elements are expected to occur (that is, "when" information; e.g., word initial/final positions); and iii) the interplay between ventral and dorsal attention networks will be particularly critical in the generalization of learned rules to new instances in a language with analogous dependencies between elements. More specifically, we predicted that goal-directed attention to the time-positions of rule elements will facilitate the fast learning of new dependencies by the binding of new "what" (stimulus identity) and "when" (stimulus temporal position) information [32].

## Results

In order to test our hypotheses, we followed the protocol illustrated in Fig 1. Participants were exposed to an artificial language in 2 different sessions: Session 1 tested rule learning, and Session 2 tested rule generalization. In all cases, the artificial language/s used comprised 3-word phrases with no semantic content and an embedded rule consisting of a dependency between the initial and final elements of a phrase. Importantly, the results obtained with this kind of stimulus material are similar to those observed with real language rules [33,34] (e.g., *is* sing*ing*, *is* play*ing*).

Session 1 consisted of 2 parts: in Part 1, all participants were exposed to an artificial language (L1) in an incidental learning task. In Part 2, participants were re-exposed to the same (L1) language and task but divided into 2 groups depending on whether they were functional MRI (fMRI) scanned (intervention group, N = 22) or performed the task outside of the scanner (control group, N = 32). The control group was included to assess task repetition effects.

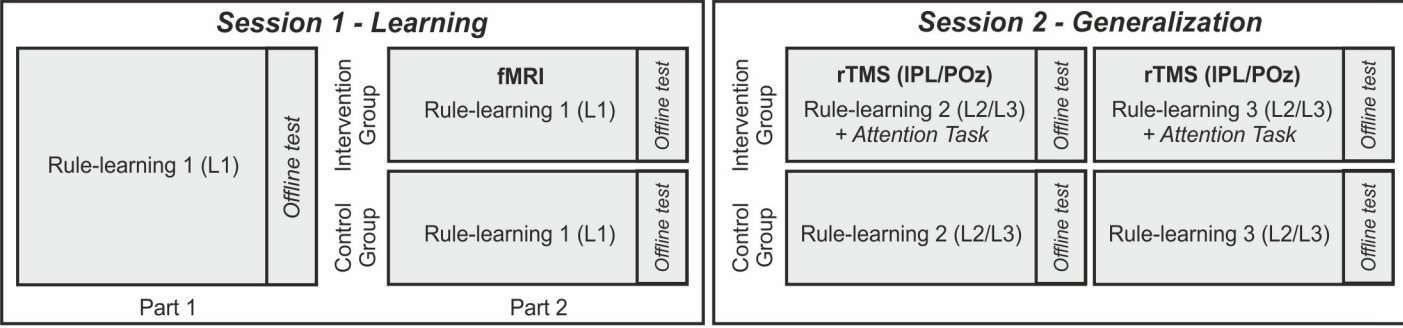

**Fig 1. Schematic overview of the protocol.** Sessions 1 and 2 were conducted a week apart. Session 1 assessed learning using a single language. In Session 2, generalization was assessed with 2 new languages that followed an analogous structure to the language learned in Session 1. Languages were counterbalanced across the protocol. rTMS intervention order (lPL/POz) was counterbalanced between participants in the intervention group and performed on the same day. The control group followed the exact same protocol but had no fMRI or rTMS intervention. L1, L2, L3 = Languages 1–3. fMRI, functional MRI; lPL, left parietal lobe; POz, midline posterior location according to the 10–20 system electrode location; rTMS, repetitive transcranial magnetic stimulation.

Session 2 was conducted 1 week after the initial session. This session was designed to test participants' ability to generalize the rule learned on Session 1 to new but structurally analogous artificial languages (L2 and L3; that is, L2/L3 followed the same structure as L1 but instantiated in totally new words). In Session 2, participants in the intervention group (top row, Fig 1) received repetitive transcranial stimulation rTMS) on the peak of activation within the left parietal lobe (lPL) and on a control site (POz [midline posterior location according to the 10–20 system electrode location]), successively. Participants learned the new languages under the effects of the stimulation on each site. The order of stimulation was counterbalanced across participants. The hotspot in the lPL was individually identified from the fMRI scans of the second part of Session 1. We predicted this to reflect goal-directed attention engaged at a later stage of learning, following Part 1. The lPL area targeted in the rTMS intervention was chosen a priori for its known role in temporal attention [27,35], which was hypothesized to play a crucial role in the generalization of linguistic rules. Targeting the lPL with rTMS in Session 2 was thus expected to hinder participants' goal-directed attention and hence the generalization of the rule learned in Session 1. The control group (bottom row, Fig 1) was tested with the same language tasks and with the same delays and order as the intervention group but without receiving rTMS. The performance of the control group additionally informed us about the expected effects of language generalization without rTMS.

In each session, we evaluated rule learning of nonadjacent dependencies in artificial languages [16]. We used 2 types of blocks. In rule blocks, participants were exposed to 3-word phrases containing AXC-type dependencies, with the first word A predicting the last word C irrespective of the intermediate word X (e.g., "jupo [variable word] runi"). In no-rule blocks, 3-word phrases with no dependencies (XXX-type and XXC-type strings) were used. Each 3-word phrase was considered a trial. Rule and no-rule trial blocks were interleaved (see Materials and methods). Participants were not informed about the presence of rules; their task was to detect the presence or absence of a given target word (e.g., "runi") that always took the third position in the phrase in both rule and no-rule blocks.

The word-monitoring task measures incidental learning of the dependencies [16] such that, as learning progresses, participants are expected to start anticipating the presence/absence of the target word (C) upon hearing the first word (A) of the phrase in the rule blocks. Learning progression, evidenced by a reaction time (RT) gain over trials, can be approximated as a learning slope via regression analysis (Fig 2). A negative learning slope can thus be interpreted

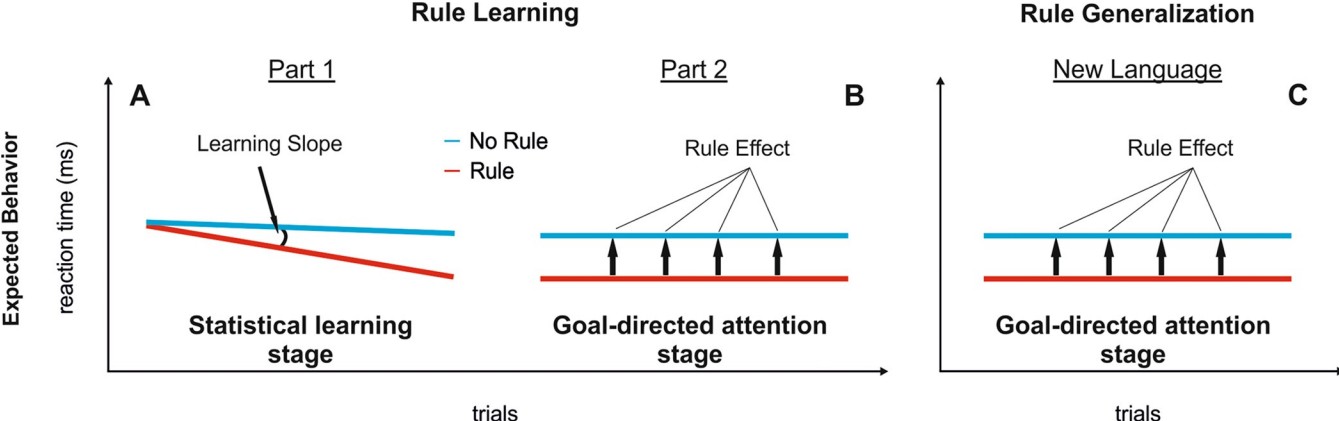

**Fig 2. Hypothesized RT slopes for rule and no-rule blocks over repetitions of the incidental rule-learning task.** For each artificial language learned, participants were exposed to blocks with rules, in which the initial word determined the identity of the last word of the phrase, and no-rule blocks, in which the final word could not be predicted based on the first one. (A) Part 1: Reflecting statistical learning, rule blocks are expected to exhibit a greater gain in RTs across trials than no-rule blocks as a consequence of the ability to predict the upcoming occurrence or absence of the target word. The difference between rule and no-rule slopes (learning slope) is thus a measure of statistical learning indicating progressive rule learning in the early stages. (B) Part 2: If participants can benefit from previous learnings to orient attention to the initial element to consistently anticipate the final one, their RTs for rule blocks should plateau in later learning stages and show a sustained difference compared to no-rule blocks throughout (rule effect, that is, the mean difference in RT between rule and no-rule trials). (C) A plateau should also be observed for participants that generalize their attentional focus on initial and final elements to a new language with the same type of dependencies (that is, rule). RT, reaction time.

as an indication of statistical learning in the initial stages (Fig 2; see also Materials and methods). As learning accrues, we expected participants' RTs to gradually reach a plateau, indicating that they can now use the initial element of each dependency to consistently anticipate the dependent element. Such plateaux should be expressed in later learning stages as stable RTs over the course of the trials (relatively flat learning slope) for the rule blocks, that is, a slope similar to no-rule blocks albeit with faster mean RTs in rule blocks compared to no-rule blocks. This RT advantage in rule blocks (henceforth "rule effect") is calculated as the difference in the mean RT between rule and no-rule blocks; Fig 2).

This pattern of behavior is to be expected if participants can apply the knowledge acquired during the initial stage to focus their attention in a goal-directed manner on the initial element of the phrase in order to consistently predict the appearance of the target word at the end of the phrase. Finally, a stable rule effect (that is, a sustained difference in RT between rule and no-rule phrases) should also be expected in the learning of a new but structurally analogous language, given by the generalization of the learned attentional focus to the new material [20,21].

## Signatures of statistical learning and attention-based behavior

During the first session, participants (N = 54, 39 women, mean age = 22.61 years, SD = 5.75) performed the incidental rule-learning task on an artificial language (e.g., L1). A short break was provided midway through the session (Session 1; see Fig 1). Out of the main cohort, a subgroup of participants (intervention group; N = 22, 13 women, mean age = 23.63 years, SD = 4.67) was randomly selected to undergo the rTMS intervention in Session 2. In order to obtain the appropriate coordinates for rTMS stimulation, participants in the intervention group performed the second part of the rule-learning task in the MRI scanner (that is, rule-learning 1, Part 2). To assess task repetition effects, the remaining participants (control group;

N = 32, 26 women, mean age = 21.9 years, SD = 6.37) were behaviorally tested with the exact same protocol as the intervention group but without fMRI in Session 1 or rTMS in Session 2.

Fig 3 illustrates changes in RT across trials as the task was repeated across the 2 parts of Session 1. A linear mixed model approach was used to obtain the learning slopes for the 2 rule-learning parts of Session 1 and to compare rule versus no-rule performance within each part of the session. In addition, rule effects were calculated by contrasting mean RTs in rule and no-rule blocks within each part and were assessed via paired $t$ tests (see Materials and methods; 4 subjects were excluded from Part 1 because of missing data).

For the entire sample, a steeper learning slope (that is, more negative) for rule blocks compared with no-rule blocks was observed during Part 1 ($\beta_{diff}$ = −0.73, $t$ = −3.86 $p < 0.0002$; $\beta_{diff}$ is the estimate of the difference in learning slopes between the rule and no-rule conditions, see Materials and methods). The significant difference indicates that learning of the statistical relations between words occurred at this stage. As expected (Fig 2), nonsignificant (flatter) slopes for rule blocks were then obtained the second time the task was performed (Part 2: $\beta_{diff}$ = −0.36, $t$ = −1.81, $p > 0.07$). Crucially, the slope in the rule condition in Part 1 was significantly more pronounced than that of Part 2 ($\beta_{diff}$ = 0.69, $t$ = 3.32, $p < 0.001$). These changes in the slope were accompanied by a significant rule effect in both Part 1 ($t[49]$ = 3.63, $p < 0.001$, $d_{Cohen}$ = 0.513) and Part 2 ($t[53]$ = 4.00, $p < 0.001$, $d_{Cohen}$ = 0.545). The rule effect was equivalent in Parts 1 and 2 (nonsignificant difference between the two, $p = 0.649$, $d_{Cohen}$ = 0.065). The combined findings of a nonsignificant learning slope (significantly flatter than that for Part 1) and a significant rule effect in Part 2 supports the view of rule learning as a two-stage process

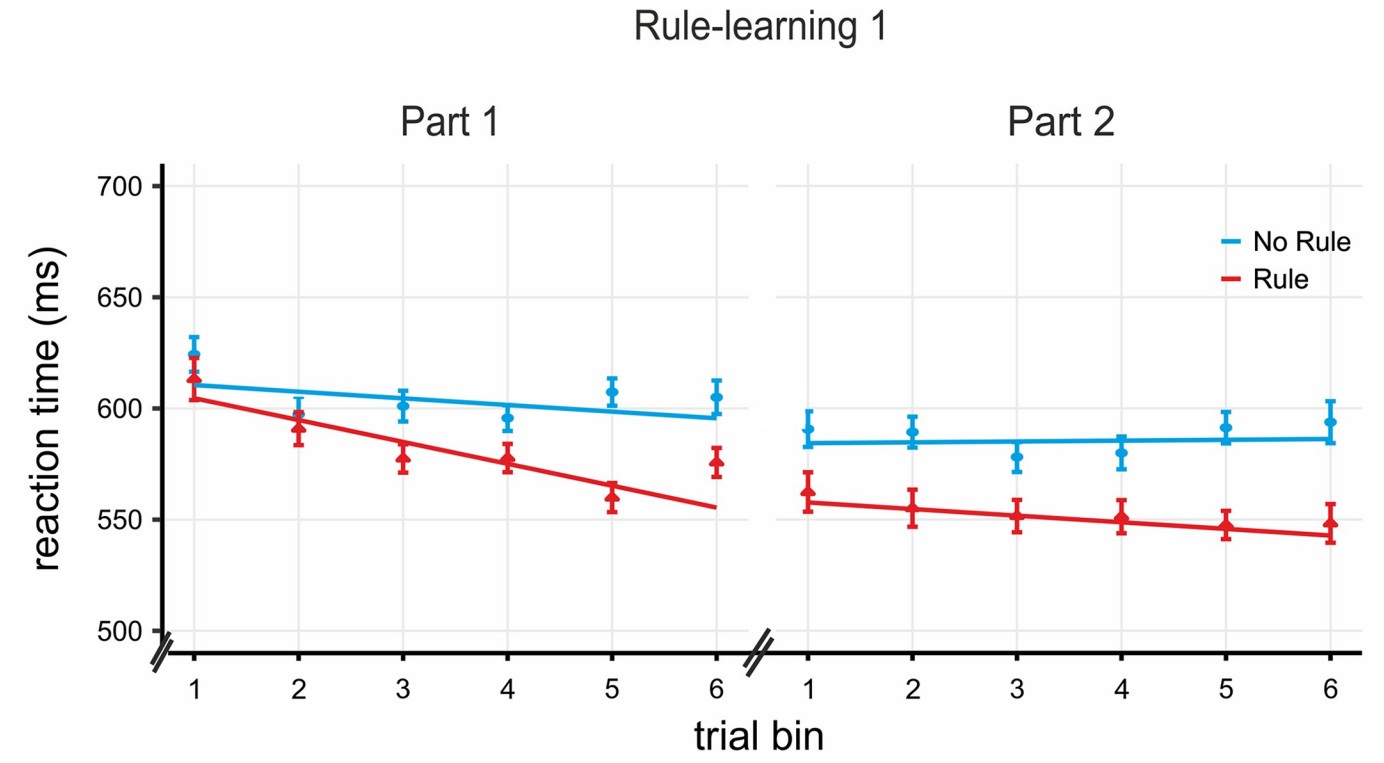

**Fig 3. Incidental rule-learning task results for Session 1.** Slopes for rule and no-rule blocks (N = 54) over task repetitions derived from the mixed model analysis conform to the expected pattern (Fig 2), with a significant learning slope in Part 1 and a significant rule effect with a flat (that is, nonsignificant) learning slope in Part 2. Actual data shown averaged into 6 trial bins (for visualization purposes only; the analysis did not bin the data) with the SEM over the slopes for rule and no rule derived from the mixed model analysis. Data used to generate Fig 3 can be found in S1 Data.

(Fig 2). An analysis of the intervention and control groups separately indicated a similar pattern of results in the 2 groups, except that in the intervention group, some statistical learning was still in progress in Part 2 (S1 Fig and S2 Fig).

## Functional localization of parietal areas for stimulation

With the purpose of selecting the coordinates for the rTMS intervention, we identified parietal regions engaged in the late stages of learning of Session 1 at the individual level. In particular, we localized the peak voxel with the highest activity from single-subject contrast maps for the rule versus no-rule contrast within the lPL. The decision to stimulate the lPL was taken given the critical role of this region in attention in the temporal domain [27]. As mentioned in the introduction, we expected this process to be relevant for rule learning in later stages (Part 2, Session 1; see Fig 2) as well as during generalization (that is, rule learning Part 2 and beyond). In the few cases in which the rule versus no-rule contrast showed no significant activity difference (uncorrected $p < 0.05$), we used the contrast rule condition against the implicit baseline (N = 5). In order to illustrate the variability in individual activation during rule learning, masks for each participant's activation pattern were calculated using a $p < 0.005$ uncorrected threshold as a cutoff. These individual masks were then added to create a group overlap map. S3 Fig depicts the resulting overlap, showing voxels that were activated in at least 10 participants at the individual level during the rule-learning task. Two subjects were excluded from this analysis because of technical failure during fMRI scanning.

## Exploration of the brain networks engaged during late learning stages (Part 2, Session 1)

Given the potential mixture of slow and fast learners in the intervention group (S2 Fig), we performed an exploratory individual differences analysis of their fMRI activity. This analysis was aimed at investigating the networks related to statistical learning and goal-directed attention by correlating participants' corresponding behavioral measures (that is, learning slope and rule effects) with the appropriate fMRI contrasts.

To identify the neural activity related to statistical learning, we derived individual learning slopes for rule blocks ran in the fMRI (Part 2, Session 1). These individual learning slopes were then correlated with the blood-oxygenation-level–dependent (BOLD) signal change in the same blocks (see Materials and methods). Covarying with individual differences in the slope for rule blocks, this analysis revealed a ventral frontoparietal network as related to statistical learning (see Fig 4A). This ventral network included 2 large clusters of perisylvian areas in the left hemisphere, including the left IFG and the left iPL, and a smaller cluster centered on the right IFG (S1 Table).

Because we hypothesized that goal-directed attention would increase over exposure to the same artificial language, to identify neural activity related to this process, we estimated individual measures corresponding to the change in the rule effect from Part 1 to Part 2 (that is, Part 2 rule effect minus Part 1 rule effect). We then correlated this rule effect increment with the rule versus no-rule fMRI contrast. Individual differences in the increment of the rule effect from Part 1 to Part 2 covaried with activity in a more dorsal and bilateral frontoparietal network, consistent with the engagement of goal-directed attention (see Fig 4B). This dorsal network included 2 large clusters centered on bilateral and predominantly superior parietal regions extending to the precuneus, as well as left middle and superior frontal gyri. In addition, there was also an involvement of the left iPL and bilateral basal ganglia (S2 Table).

Note that because of the limited number of subjects, these results can only be interpreted tentatively. However, they do hint in the direction of the hypothesized gradual transition from

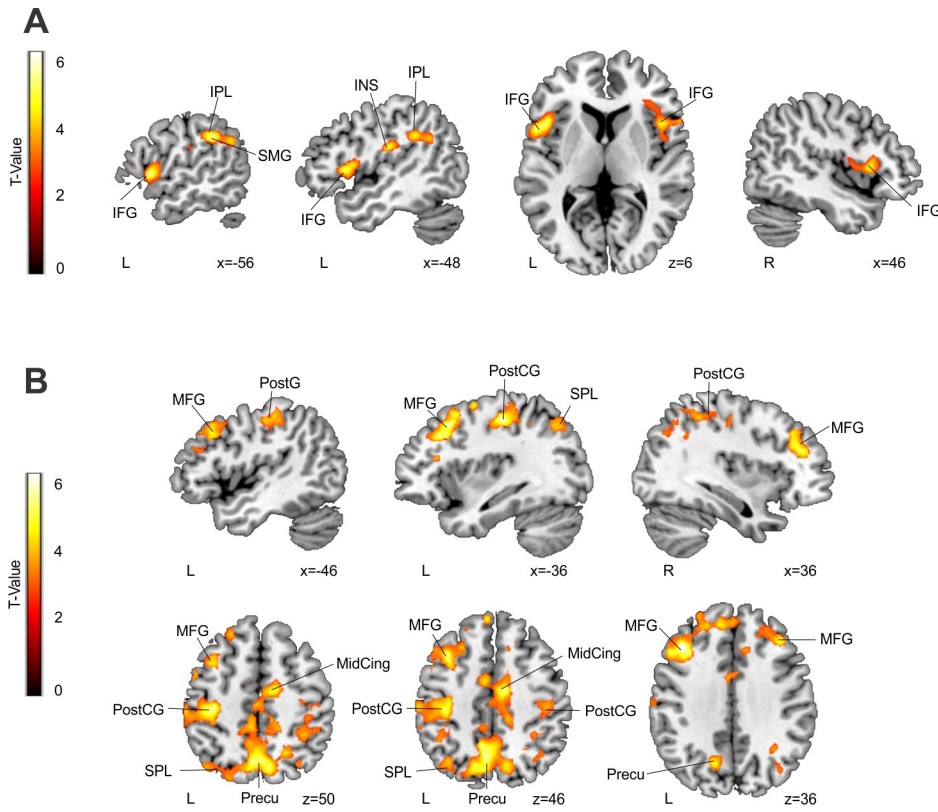

**Fig 4. (A) BOLD signal over a ventral frontoparietal network in rule blocks significantly covaries with their corresponding measure of statistical learning (learning slope); the more activity, the greater (that is, more negative) the slope (see also S4 Fig).** (B) BOLD signal activity over a dorsal frontoparietal network (rule blocks minus no-rule blocks) covarying with the measure of goal-directed attention (Part 2 rule effect minus Part 1 rule effect); the more activity, the larger the effect. Only significant results ($p < 0.05$ FWE-corrected at the cluster level, with an additional $p < 0.005$ at the voxel level and 50 voxels of cluster extent) are shown for both analyses. Neurological convention is used with MNI coordinates shown at the bottom right of each slice. https://identifiers.org/neurovault. collection:8592. BOLD, blood-oxygenation-levelamily-wise error; IFG, inferior frontal gyrus; INS, insula; iPL, inferior parietal lobule; MFG, middle frontal gyrus; MidCing, midcingulum; MNI, Montreal Neurological Institute; PostCG, postcentral gyrus; Precu, precuneus; SMG, supramarginal gyrus; SPL, superior parietal lobe.

more ventral frontoparietal regions for statistical learning to more dorsal frontoparietal areas for goal-directed attention in later stages of learning.

## Is the lPL causally related to rule generalization and goal-directed attention?

The same participants (intervention and control groups) came to a second session 1 week later (Session 2; Fig 1) to test whether the lPL areas activated in the late stages of learning (functionally localized in Session 1) were causally related to rule generalization and whether this area was indeed involved in goal-directed attention for the binding of "what" and "when" information. Session 2 thus assessed participants' rule-learning performance on 2 new languages (L2 and L3) that followed the same rule structure as the language (L1) used in Session 1 (that is, in which the initial word determined the identity of the final word). Participants in the intervention group performed the rule-learning task after rTMS on the lPL (individually determined by the functional localization analysis; see Materials and methods) and a control site (POz; order of stimulation site and language counterbalanced). No systematic activation at the POz

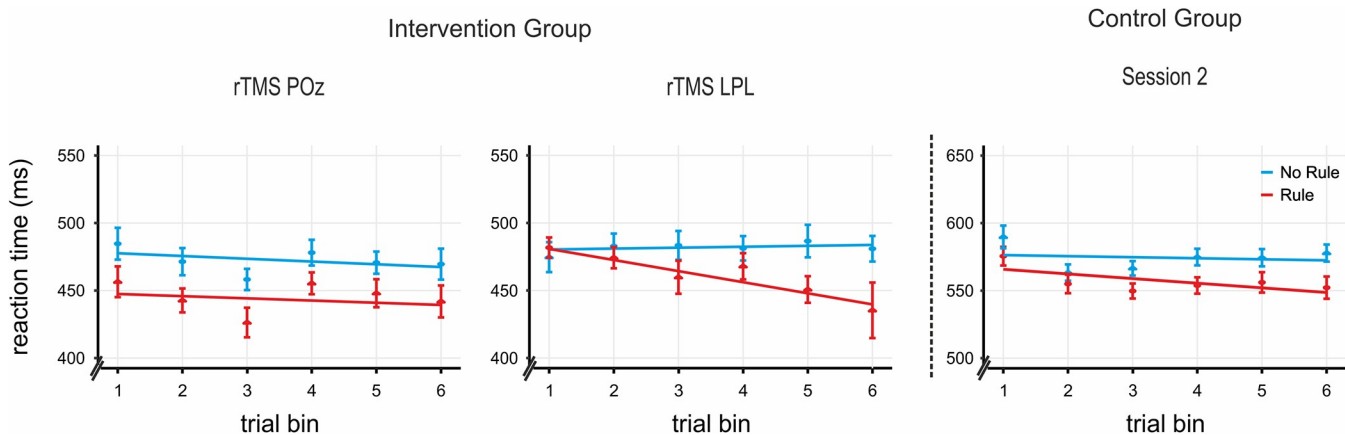

**Fig 5. Incidental rule-learning task results for Session 2.** Both rTMS POz intervention (left panel) and control group Session 2 (right panel) show the expected pattern of rule learning with a significant rule effect and a nonsignificant learning slope, indicating that attentional focus generalized to the learning of the new languages. In contrast, a significant learning slope under rTMS lPL effects (center panel) suggests a return to the progressive rule learning of early learning stages (that is, statistical learning). Actual data shown averaged into 6 trial bins (for visual purposes only; the analysis did not bin the data) with the SEM over the slopes for rule and no rule derived from the mixed model analysis. Data used to generate Fig 5 can be found in S2 Data. lPL, left parietal lobe; POz, midline posterior location according to the 10–20 system electrode location; rTMS, repetitive transcranial magnetic stimulation.

location was observed in the individual contrasts (see S3 Fig) or in the regression analyses (see Fig 4, S1 Table and S2 Table), confirming the appropriateness of this location as a control site. Participants in the control group were also tested in Session 2 with the same new languages (L2 and L3), just like the intervention group, but without rTMS (control group Session 2; see Fig 1). Their performance was used to measure the effects of task repetition and to have a measure of the baseline levels of generalization without rTMS.

We expected participants' goal-directed attention to be engaged in the transfer of the rule knowledge acquired in Session 1 to structurally similar languages in the rule-learning tasks of Session 2. A mixed model analysis of their RTs indicated, as predicted, that rule performance in the control group in Session 2 followed the same pattern observed in Part 2 of Session 1 despite performing the task on a new language. That is, participants showed faster RTs in rule blocks compared with no-rule blocks (rule effect: $t[31] = 2.434$, $p < 0.021$, $d_{Cohen} = 0.430$) but no significant slope difference between the blocks ($t[31] = -1.6$, $p > 0.11$; Fig 5). Moreover, the slope for the control group during Session 2 was not significantly different from their slope for Part 2 in Session 1 ($t[31] = -0.84$, $p > 0.4$).

The same pattern was observed for the intervention group after rTMS to the control site POz (rule effect: $t[19] = 2.85$, $p < 0.01$, $d_{Cohen} = 0.638$; rule versus no-rule slope difference: $t[19] = 0.182$, $p > 0.85$). Indeed, both the rule effect and the rule slope under rTMS POz were comparable with those of the control group (respectively, $t[50] = 2.85$, $p = 0.271$, $d_{Cohen} = 0.317$ and $t[51] = -0.79$, $p > 0.4$). This indicates that stimulation to POz did not influence task performance (that is, the control group and the intervention group during rTMS POz have comparable slopes and rule effects) and thus confirms rTMS POz as a suitable control site for participants' performance under rTMS lPL. This pattern of results supports the idea that participants' knowledge of the rule structure acquired early on in learning (Session 1) was used to perform the task in this later phase. Thus, the behavioral plateau achieved in the later stages of Session 1 (Part 2) was maintained when a new language following the same type of rule was being learned and could be used in generalization.

Finally, we assessed the effects of rTMS over lPL on rule generalization performance. As can be observed in Fig 5, the pattern of performance in the intervention group after the

application of rTMS on the lPL stood in stark contrast to the pattern seen in both the control group and in the intervention group after rTMS to the control site POz (Session 2). First, the slope difference for rule blocks compared to no-rule blocks was significant after rTMS stimulation to the lPL ($\beta_{\text{diff}} = -0.95$, $t = -3.75$, $p < 0.0002$; Fig 5), in line with the pattern observed in the early stages of rule learning (that is, Session 1, Part 1). The slope for rule blocks in this phase was significantly more negative than both the slope after rTMS stimulation to the control site POz ($\beta_{\text{diff}} = -0.67$, $t = -2.66$, $p = 0.008$) and that of the control group Session 2 ($\beta_{\text{diff}} = -0.57$, $t = -2.49$, $p = 0.013$), suggesting that after rTMS over the lPL, subjects performed the task capitalizing on statistical learning. Note that the effects of rTMS on the lPL were not driven by the performance in the no-rule blocks; there were no significant differences between the slopes for the no-rule blocks in rTMS lPL and rTMS POz ($\beta_{\text{diff}} = 0.29$, $t = 1.17$, $p = 0.24$), rTMS POz and control group Session 2 ($\beta_{\text{diff}} = 0.21$, $t = 0.94$, $p = 0.35$), or rTMS lPL and control group Session 2 ($\beta_{\text{diff}} = -0.08$, $t = -0.33$, $p = 0.74$). To rule out any order effects (lPL stimulated first/second), order was added as a factor to the model for both rTMS lPL and rTMS POz phases (Materials and methods), yielding nonsignificant effects in both cases ($p > 0.1$). Finally, we note that the overall faster RTs for rTMS conditions compared with RTs for the control group may be due to nonspecific rTMS effects [36] and/or the lower latency equipment employed for testing in the rTMS facilities and do not affect the main outcome.

### The effects of rTMS on lPL on goal-oriented attention

To discern the precise role of the lPL, participants in the intervention group performed an additional attention task specifically designed to assess the ability to use content ("what": e.g., "un" predicting "able" in **un**forgett**able**) and temporal cues ("when": e.g., word initial/final positions). The attention task was administered under the effects of rTMS to lPL and POz, just like the rule-learning task (Fig 1), with the order of tasks counterbalanced between participants (Fig 6). Participants were asked to judge whether the pitch of a target syllable at the end of a

**Fig 6. Illustration of the experimental design of the attention task.** Participants had to judge the pitch of a target syllable presented after a sequence of alternating syllables. The pitch of the target syllable could be either higher or lower than that of the preceding sequence of syllables. Sequences of syllables were presented either rhythmically to engage temporal orienting (attention to "when") or nonrhythmically (50%), with an otherwise constant trial length. At the same time, the initial syllable of each sequence could be informative or noninformative of the identity of the target syllable (50%), manipulating identity-based attention (that is, attention to "what"). ISI, interstimulus interval.

sequence of alternating syllables was higher or lower than the preceding sequence. In order to build a task as similar as possible to the language paradigm used for the rule-learning experiment, we used syllables as the auditory stimuli on which to manipulate attention. More critically, in the language task, the temporal interval between words was fully predictable. Thus, for the attention task, temporal orienting ("when" information) was manipulated by presenting the sequences of syllables in either rhythmic (with isochronous interstimulus intervals) or nonrhythmic form (variable time interval between syllables) while keeping the overall trial length constant. Therefore, in both tasks, the temporal relation between elements follows the same implicit temporal expectation. We chose rhythm because the empirical and theoretical foundations of isochronous-anisochronous event timing are well-established as a manipulation of temporal attention [35,37,38]. In addition, there is solid evidence with respect to the effect of rhythm in speech and rule-dependency processing and learning in language and an extensive theoretical background relating temporal attention to speech processing [39–43]. On the other hand, in the language task, the identity of the first element informed the identity of the last element in rule blocks, whereas this was uninformative in the no-rule blocks. Thus, in the attention task, the first syllable of the sequence was informative or uninformative of the identity of the final target syllable. Attention to the content ("what" information) was manipulated by asking the participants to focus their attention to the identity of the first syllable because this would help them to perform the task. The attention manipulations used and the orthogonality of the participant's goal (that is, pitch discrimination) are comparable with those previously used in temporal attention research [38,44,45]. Participants were familiarized with the attention task before receiving rTMS.

In order to test whether "what" and "when" cues were used differently in the 2 intervention conditions, we performed a $2 \times 2 \times 2$ repeated-measures ANOVA on RTs to the target syllable (correct responses only), with intervention (rTMS_lPL, rTMS_POz), rhythm (rhythmic, nonrhythmic) and identity (informative, noninformative) as within-subject factors. This analysis revealed a main effect of intervention ($F[1,18] = 13.36$, $p = 0.002$, partial $\eta^2 = 0.426$), with overall RTs slower after rTMS to the lPL compared to rTMS POz. Regarding the use of "when" cues, pitch discrimination in rhythmic sequences was faster than in nonrhythmic sequences (main effect of rhythm: $F[1,18] = 24.62$, $p < 0.001$). Concerning the effect of "what" cues, there was also a facilitation in RTs for informative compared with noninformative sequences (main effect of identity: $F[1,18] = 9.01$, $p = 0.008$). The main effects of rhythm and identity thus indicated that the attentional manipulations were effective, and participants benefited from the use of both "what" and "when" information. Crucially for our hypothesis, we observed a significant triple interaction between intervention, identity, and rhythm ($F[1,18] = 7.54$, $p = 0.013$, partial $\eta^2 = 0.295$), not affected by the order of rTMS intervention ($F < 1$), suggesting differences in how "when" and "what" cues were used under each rTMS intervention (Fig 7). We then proceeded to unpack the interaction by analyzing how information about "what" was affected by the presence or absence of predictive information about "when" the target is expected in each rTMS intervention. In sequences in which "when" information was unpredictable (nonrhythmic sequences), Identity had a comparable facilitatory effect in both rTMS sites (identity: $F[1,19] = 9.58$, $p = 0.006$, partial $\eta^2 = 0.335$; intervention $\times$ identity: $F[1,19] = 1.12$, $p > 0.3$, partial $\eta^2 = 0.056$). However, in sequences in which "when" information was predictive (rhythmic sequences), identity had a facilitatory effect (identity: $F[1,19] = 4.64$, $p = 0.044$, partial $\eta^2 = 0.196$) that differed in the 2 rTMS-stimulated sites (intervention $\times$ identity: $F[1,19] = 13.18$, $p < 0.002$, partial $\eta^2 = 0.410$). Although participants could benefit from knowing "what" the identity of the target was while stimulated at the control site (rTMS POz, $t[19] = 5.51$, $p < 0.0001$, $d_{Cohen} = 0.596$), this facilitation disappeared under rTMS lPL stimulation ($t[19] = -0.22$, $p > 0.8$, $d_{Cohen} = 0.027$). These results indicate

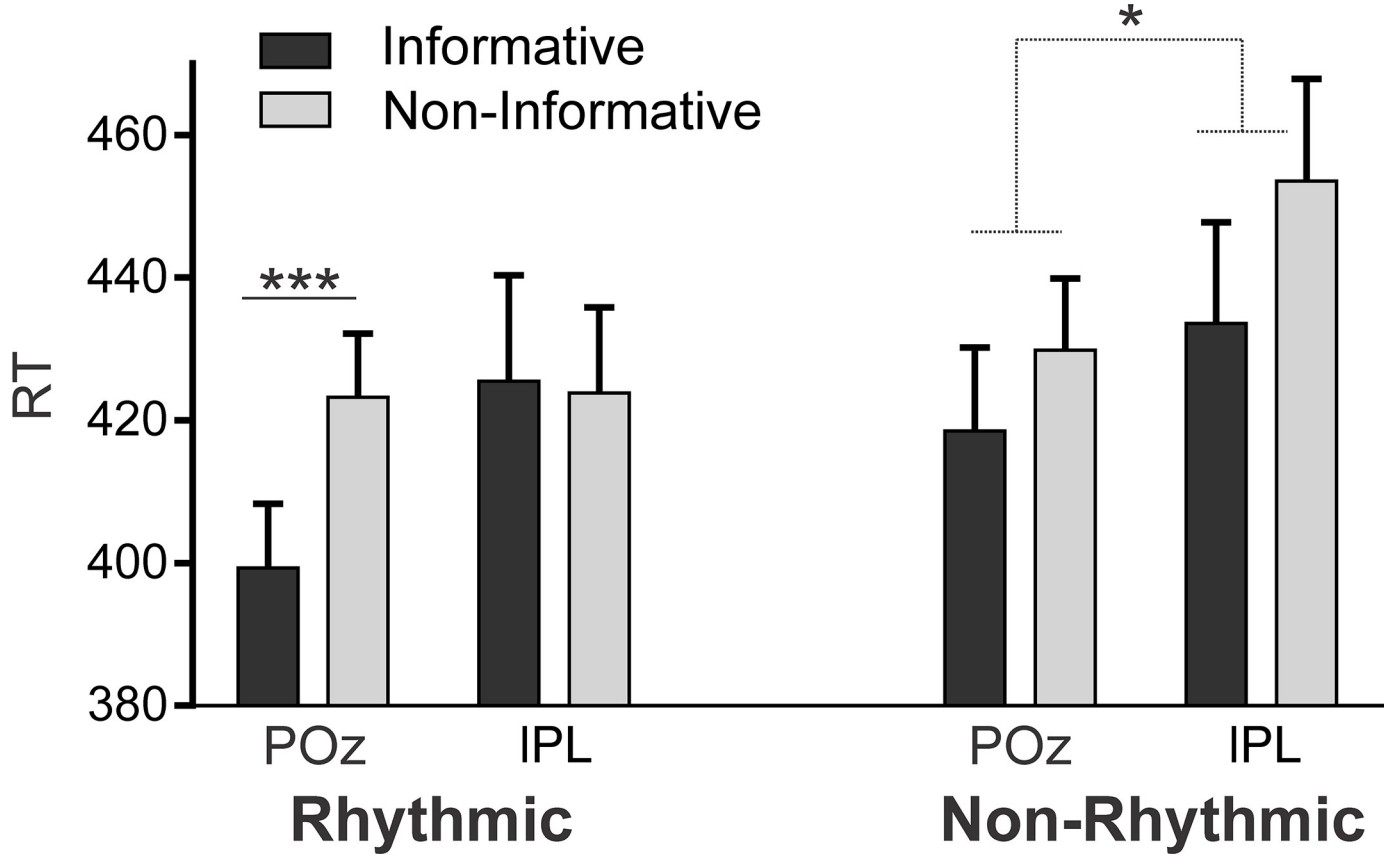

**Fig 7. Attention Task results.** Black and gray shaded bars represent mean RTs with their SEM for informative and noninformative trials, respectively, for the phases with rTMS on POz and lPL, separated into rhythmic and nonrhythmic conditions (***$p < 0.0001$, pairwise comparison; main effect of identity; *$p < 0.006$). Data used to generate Fig 7 can be found in S3 Data. lPL, left parietal lobe; POz, midline posterior location according to the 10–20 system electrode location; RT, reaction time; rTMS, repetitive transcranial magnetic stimulation.

that although able to benefit from the "what" information in isolation, participants were impaired in their ability to benefit from it when both "what" and "when" cues were presented in conjunction.

The rTMS lPL intervention therefore appeared to impact the interaction between the 2 forms of attention, that is, on the joint use of both "when" and "what" cues to optimize performance in the pitch discrimination task. In sum, although rhythm and information about future content seem to act jointly to decrease RTs for discrimination, this benefit was lost under rTMS at lPL, in which either content or rhythm, but not both concurrently, could be used to improve task performance.

## Discussion

In this study, we provide converging behavioral and neuroimaging data in favor of the hypothesis that successful rule learning is a two-stage process. In the early exposure to an artificial language, participants showed a gradual decrease in response times for rule compared with no-rule phrases, consistent with the progressive learning of the statistical relationship between rule elements (Fig 3, left panel). In the latter part of exposure, a sustained advantage in response times for rule phrases over no-rule phrases (Fig 3, right panel) is supported by a dorsal frontoparietal network (Fig 4 and S3 Fig). This advantage in response times was maintained

for new languages with the same kind of rule. This is consistent with the hypothesis that goal-directed attention in later learning stages facilitates the abstraction of the underlying rule, allowing participants to generalize their acquired knowledge regarding the position (that is, initial/final) of the elements of the rule to new material (Fig 5, control group and intervention group, left panel). Moreover, rTMS to a key node of the dorsal network (the lPL) interfered with rule generalization (Fig 5, intervention group, right panel), pointing to a further role for goal-directed attention in this stage. Finally, by means of an additional attention task, we were also able to specify the mechanism underlying generalization as carried out by the dorsal network. In particular, rTMS to the lPL hindered participants' ability to integrate stimulus temporal and identity information, that is, information about "what" element is expected and "when" (Fig 7), suggesting that rule generalization involves the integration of such features.

Here, by contrasting RTs over trials for rule and no-rule phrases in an orthogonal word-monitoring task [16], we show that rule learning is a two-stage process involving i) the progressive learning of statistical contingencies, followed by ii) the engagement of goal-directed attention for rule extraction and generalization. The use of this online measure of learning thus enabled us to tap into processes that parallel those arising in natural language learning, in which rule knowledge accrues from successive encounters with a particular grammatical rule. Our results—replicated across different sessions and cohorts—further speak for the reliability of this measure of online learning and add substantially to the knowledge gained by the use of test measures acquired after language exposure (that is, offline), which are blind to the learning dynamics [46,47].

We hypothesize that, although still tied to specific content (e.g., the particular words forming a dependency), the ability to direct attention to particular moments in time at later learning stages leads to the emergence of abstract knowledge regarding classes of words (that is, categories of words that can occur at specific positions in a phrase), which facilitates generalization (cf. [48]). Note that category knowledge implies the notion of time-bound category slots and placeholders relating to how specific content is apt to take a particular slot in time. What is further required for generalization is the transfer of the learned temporal structure (that is, the attentional focus) to the new material as it occurs in the acquisition of a second language [20]. That is, generalization simply involves the reverse process of filling in the category slots with the specific word-forms of the new language.

The two-stage account of rule learning evidenced at the behavioral level is also supported by the observed shift in the cortical networks involved. Approximated as a learning slope, statistical learning in the early stages correlated with brain activity in a left-lateralized ventral frontoparietal network (Fig 4A). In contrast, later-arising sustained rule effects (mean RT differences between rule and no-rule phrases) engaged a more dorsal and bilateral frontoparietal network (Fig 4B) related to goal-directed attention [23]. Although these fMRI analyses should be taken with caution because of the limited sample size, activity in the reported areas is consistent with the existing literature on both language learning [24,26,27,34–38] and attention [23].

Regarding language learning, several studies have previously related regions of the ventral network, such as the IFG and iPL, to statistical word [51–53] and rule learning [13,28,30,31,49,50,54,55]. These studies have significantly added to a wealth of literature implicating the IFG in syntax processing more generally (e.g., [29,56–58]) by extending its role to the learning of at least simple syntactic rules. Opitz and Friederici, e.g., reported an increase in the activity of ventral premotor cortex following a rule violation during the learning of an artificial grammar [28]. More anterior areas in the IFG have also been related to successful nonadjacent dependency learning more specifically [59], along with the caudate nucleus of the basal ganglia [31] also observed in our fMRI results. Goranskaya and colleagues additionally found

that nonadjacent dependency learners activated a broader frontoparietal network, including not only left frontal but also bilateral parietal regions [30]. By bridging rule learning and attention, our study now sheds light onto the precise role of these regions. Indeed, the same frontoparietal network has been classically associated with the detection of behaviorally relevant stimuli (that is, stimulus-driven attention [24]). More recently, however, this ventral attention network has also been shown to respond to statistical regularities in the environment (cue–target contingencies [25]). This may be particularly interesting for the interpretation of our results because the engagement of the ventral attention network in our work could reflect the detection of rule elements (predictive and predicted), which become relevant by virtue of their statistical relationship. This link between the learning of statistical dependencies and stimulus-driven attention is in line with previous studies relating language rule learning to an event-related potential (the P2) associated with this form of attention [13,54,55]. Finally, a left-lateralization of the ventral network has been shown for stimulus-driven temporal orienting [27]. This is consistent with our reported left-lateralized activity in this network in response to rules instantiated in speech, which necessarily imposes a temporal dimension.

In contrast to the ventral network, the increment in rule effect arising in later learning stages was related to activity in a more dorsal and bilateral frontoparietal network. This dorsal network has been traditionally associated with the goal-directed orienting of attention in both the spatial [23] and temporal [27] domains. We further confirmed the causal implication of this network in rule generalization by the interference of its left parietal node, identified for each individual during later learning. Indeed, rTMS to this region hindered participants' ability to use acquired rule knowledge for the fast learning of new dependencies. It is important to mention though that the causal role of this region does not imply that other brain areas within this network, including the contralateral parietal lobe, do not play a role. Further research is necessary to answer this question. It is interesting as well that whereas the same participants stimulated on a nonrelevant cortical area (POz) showed the expected generalization behavior, in line with the control group, participants under the effects of rTMS lPL were seen to return to progressive rule learning, suggesting that left parietal stimulation did not hinder statistical learning. In other words, rTMS on the lPL did not impede the computation and use of statistical and temporal order information for the gradual learning of the rules. This further speaks for the relative independence of the 2 mechanisms, congruent with their putative support by the 2 reported cortical networks.

Finally, we devised an attention task to clarify the mechanisms underlying the lPL function. In particular, we measured participants' ability to use temporal and stimulus identity cues (that is, information about "when" and "what," respectively) under the same rTMS effects. Our results show that although participants could still benefit from either "what"- or "when"-type cues, rTMS on the left parietal cortex interfered with the use of both kinds of cues in conjunction. This suggests that the binding of stimulus temporal and identity information is required for successful generalization and is a characteristic of goal-directed attention in latter learning stages. We were careful to design the attention task to match the language task as closely as possible, and the results obtained support our hypothesis. The lPL was proven to be relevant for the integration of what and when information, a mechanism that we claim is key in the rule generalization process. The exact same region was critical for rule generalization. Despite this, we acknowledge that the fact that the same lPL region has both an effect on generalization and in the integration of "what" and "when" information does not necessarily imply that the 2 functions are linked. It could simply be due to the lPL supporting both functions without them being related. Further investigations will shed light onto this issue. However, it is worth noting that our proposal coincides with the role of the parietal lobe in structure learning very recently proposed in the visual domain. This function is argued to derive from this area encoding the

"relative position of objects in space" and "the relations among entities in abstract conceptual space" [60].

On a more general note, our results are consistent with the dual-process hypothesis of rule learning from speech that we recently proposed in the developmental domain [17]. Specifically, we hypothesized that the maturation of the dorsal frontoparietal network facilitates the exploitation of knowledge acquired through statistical learning via attentional mechanisms. Hence, the gradual acquisition of statistical contingencies via incidental learning will allow children after their second year of life to form abstract, long-lasting rule-based knowledge. The results of the present study suggest a similar dual-process mechanism in adult learners when facing a new language.

In conclusion, our data support the view of 2 distinct stages in rule learning, predominantly characterized by the engagement of stimulus-driven and goal-directed attention, respectively. Importantly, we have established a causal link between left parietal function as engaged in later learning stages and the abstraction and subsequent generalization of language rules. Interfering with this region caused both an impairment in goal-directed attention and in the ability to generalize rules in language. In considering the precise implication and role of attentional mechanisms, therefore, this view of rule learning integrates seemingly irreconcilable experimental findings and theoretical proposals (e.g., [8,9,11,22,61]).

## Materials and methods

### Participants

The study included one group of participants (N = 54, 39 women, mean age = 22.61, SD = 5.75) who completed the whole protocol (Session 1 and Session 2) under different conditions. A subgroup of 22 right-handed participants (intervention group: 13 women, mean age = 23.63 years, SD = 4.67) underwent an fMRI session (Part 2, Session 1) and rTMS (Session 2). The remaining 32 right-handed participants formed the control group (26 women, mean age = 21.9 years, SD = 6.37), which was exposed to the artificial languages the same number of times as the intervention group but without fMRI acquisition or rTMS. The control group was crucial to characterize the course of events under repeated exposure to the same language task and to have a baseline measure of generalization effects under no intervention. Because there was no comparable study using TMS and a similar paradigm, the sample size was determined based on previous studies applying TMS to the parietal cortex. The sample size in these studies is between 12 subjects (e.g., [62,63]) and 18 subjects (e.g., [64]), with some studies reaching 24 participants (e.g., [62,65]). Our initial sample size was set to 22, with a final sample size of 20 that was thus expected to be largely sufficient considering that we were in the range of the maximum sample size used in those TMS studies and the main comparisons were within subjects for the rTMS intervention.

All participants were native Spanish speakers and had no history of neurological or auditory problems. Participants in the intervention group were screened for compatibility with fMRI and rTMS procedures [66]. The ERC-StG-TuningLang 313841 protocol was reviewed and monitored by the European Research Council ethics monitoring office, approved by the ethics committee of the Universitat de Barcelona (IRB 00003099) and the Universitat Pompeu Fabra (CIEC Parc de Mar; 2010/3946/I), and conducted in accordance with the Declaration of Helsinki. Participants were remunerated and completed a written informed consent.

In each part and session of the experiment, several participants within each group were excluded for technical reasons or after being identified as outliers within their group. We considered participants as outliers when their learning slopes were 2 SDs above or below the mean for their group and part. In Session 1, 2 participants in Part 1 (final sample N = 30) and one

participant in Part 2 (final sample N = 31) of the control group were identified as outliers. For the intervention group in Session 1, 5 participants were excluded from Part 1 (1 outlier and 4 for technical reasons; final sample N = 17), and 2 were identified as outliers during Part 2 (fMRI; N = 19). In Session 2, none of the participants in the control group were identified as outliers (final sample N = 32). Two participants from the intervention group were excluded (final sample N = 20) because they were stimulated >2 mm from the appropriate coordinate within the lPL during rTMS (see Repetitive TMS protocol section below) because of an experimenter error.

## Experimental design

The overall experiment (main text Fig 1) comprised 2 sessions that took place approximately a week apart from one another. In Session 1, participants performed an artificial language incidental rule-learning task twice (Part 1, behavioral only for both the intervention and control groups; Part 2, behavioral for the control group and during fMRI scanning for the intervention one), followed by an offline recognition test (see S1 Text and S4 Table for further details on the offline recognition test). The artificial language (L1) and its nonadjacent dependencies were the same for both Part 1 and 2 of Session 1. For the intervention group, the fMRI data were used to identify relevant activity within the lPL during late stages of rule learning in order to select appropriate coordinates for rTMS stimulation. For the intervention group only, during Session 2, 1-Hz rTMS (15 minutes offline) was used to stimulate the individually determined maximum peak of activation within the lPL (rTMS lPL phase) or a task-irrelevant brain area (POz electrode location; rTMS POz phase). The order between the rTMS lPL phase and the rTMS POz phase was counterbalanced across participants. After rTMS application, participants performed 2 tasks: a language task similar to that in Session 1 and an attention task that tested for the effects of rTMS (lPL/POz) on the goal-directed use of content and temporal cues (that is, "what" and "when" information). Importantly, the total duration of the attention plus rule-learning tasks was kept within 30 minutes so as to ensure post-rTMS performance under the estimated rTMS effects (see section on rTMS protocol below). During Session 2, the control group completed the same language tasks as the intervention group but without rTMS stimulation.

## Rule-learning task

Three different artificial languages containing 28 bisyllabic (consonant-vowel-consonant-vowel) nonsense words each were created. One language (L1) was used in Session 1 and the remaining 2 languages (L2 and L3) in Session 2 (main text, Fig 1). The order of languages was counterbalanced between sessions and participants (see S5 Table for the full list of stimuli used).

Twenty-eight words were created for each language stream and synthesized using Mbrola speech synthesizer software [67] by concatenating diphones from the Spanish male database (https://packages.debian.org/stretch/mbrola-es2) at 16 KHz. Words (385 ms) were combined using Adobe Audition software to form 3-word phrases with 100-ms gaps between word. Phrase stimuli were presented via Presentation software (Neurobehavioral Systems) through appropriate headphones and at a volume level adjusted for the participant.

A total of 96 rule and 96 no-rule phrases (trials) were used in this task. Rule phrases conformed to an AXC structure whereby the initial word (A) always predicted the final word (C) while the middle word (X) was variable. Two different A_C dependencies (A1_C1 and A2_C2) were created out of 4 words from the total word pool. The remaining 24 served as middle (X) elements for each of the 2 A_C dependencies. The transitional probability was

always 1 between A and C elements, 0.04 between A and X, and 0.5 between X and C. Half of the no-rule trials consisted in the combination of 3 of the 24 X elements and so took the form XXX, with the only constraints that each X had an equal probability to appear in each position but could never appear twice in the same phrase. The other half of the no-rule trials consisted of the combination of 2 XX elements (following the same constraints as for XXX) followed by the participant's target word (C1 or C2). The probability of target occurrence in both the rule and the no-rule blocks was therefore 50%. Note that in the set of no-rule trials, the C element (the participant's target) occurred also in the last position, but, in contrast to the rule block, this could not be predicted on the basis of previous elements.

Participants were presented with the randomized 96 rule and 96 no-rule phrases in 4 alternated rule and no-rule blocks, with the order of blocks counterbalanced between participants. In the fMRI version of the task, data were acquired in 2 runs, including a block of rule and no rule each (counterbalanced). A short break was given between runs in the fMRI. A single offline recognition test was issued after the fourth block (see S1 Text). In order to obtain a measure of incidental rule learning, participants performed a cover word-monitoring task. Specifically, they were instructed to detect, as fast and accurately as possible via a button press, the presence or absence of a given target word, which was always one of the C elements (C1 or C2, counterbalanced). A given target word remained constant for each participant throughout the experiment and was displayed in the middle of the screen at all times for reference during the blocks. Participants were not informed about the presence of rules. Intertrial interval was jittered using pseudorandom values between 1,000 and 3,000 ms for optimal fMRI acquisition and fixed at 500 ms in the remaining phases. A maximum of 1,000 ms after the end of a given phrase was allowed for participants to respond before the next trial started. RTs were calculated from the onset the last word in the phrase until button press. Performance in interleaved rule and no-rule blocks was jointly analyzed by concatenating blocks of a same kind. Only correct response trials with RTs within mean ± 2 SDs were included for the analysis (mean over groups and conditions of 5.57% ± 3.55% of total trials removed; note that the rejection rate for the comparisons of interest was similar between groups: 4.79% ± 1.36% for the rTMS lPL phase and 4.81% ± 1.61% for the rTMS POz phase in the intervention group and 5.73% ± 3.44% for the rTMS phase in the control group).

We reasoned that if incidental rule learning occurs over exposure in the rule block, participants' gradual ability to predict the appearance or nonappearance of a target word $C_j$ on the basis of the identity of the initial word $A_j$ should be reflected in an RT gain (that is, faster RTs) over trials within the blocks (see section Linear mixed model analysis). We also expected an overall RT advantage over target words in the no-rule blocks (rule effect), in which prediction is possible (no prediction can be made during no-rule blocks). Participants' rule effect for the different parts/sessions was calculated as the mean RT difference between no-rule and rule trials.

## Linear mixed model analysis

In order to assess online rule learning in the rule-learning task within each experimental session, we used a linear mixed model approach to fit learning slopes that reflect RT gains over trials for the rule and no-rule conditions. The use of mixed models to compare the slope between conditions allows the use of RT data for all trials and subjects, which results in a more sensitive measure of the online learning process than a single mean value per subject [46,68]. Analyses were performed using the *lme4* [69] and lmerTest [70] packages as implemented in the R statistical language (R Core Team, 2012). Our basic model included RT (*rt*) and trial as continuous variables, no rule/rule (*NR_R*) and target/no target (*TNT*) as 2-level factors, and

*subject* as a factor with as many levels as participants.

$$rt \sim NR\_R + trial + NR\_R*trial + TNT + (1|subject) \tag{1}$$

$$rt = \beta0(intercept) + \beta1(NR\_R) + \beta2(trial) + \beta3(TNT) + \beta4(NR\_R*trial) \tag{2}$$

$$rt \sim TNT + (1 + NR\_R*trial|subject) \tag{3}$$

As detailed in (1), which shows the specified model, *NR_R*, *trial*, and their interaction were introduced as fixed effects terms. *TNT* was also included here albeit as a predictor of no interest. To allow for a different intercept per participant and so account for intersubject variability in basal response speed, *subject* was introduced as a random effect. The algebraic expression of the fixed effects part of the model is given in (2). Note that, in this this model, $\beta4$ (*NR_R*trial*) represents an estimate of the difference in learning slopes between the rule and no-rule conditions and can thus be used as a detrended learning slope estimate for the rule condition. For the sake of clarity, we have referred to this estimate as $\beta_{diff}$. A statistically significant negative $\beta_{diff}$ value therefore indicates that online rule learning effectively took place in that experimental phase over and beyond any reaction time gain that may be attributed to within-phase practice effects. Individual slopes were estimated for correlations via a random slopes model specifying the interaction term in the random effects part (3).

## Attention task

In addition to the rule-learning task, during Session 2, participants in the intervention group also performed an attention task with the aim of better characterizing the impact of rTMS lPL and rTMS POz. Note that all tasks were completed within 30 minutes in order to ensure post-rTMS performance under the estimated rTMS effects for both tasks. Half of the subjects performed the language task first and then the attention task, whereas the other half took the reverse order under each of the rTMS interventions (see main text Fig 1).

In the attention task (main text Fig 6), participants were asked to make a pitch discrimination judgment (higher or lower) on an isolated syllable that followed a sequence of alternating syllables. A pool of 200-ms–long syllables at 3 different pitch heights (249, 440, and 554 Hz) was created from 4 original tokens ("ba," "co," "pi," and "te") previously recorded by a female native Spanish speaker. All syllables were normalized for homogeneous output volume using Adobe Audition (Adobe Systems Software Ireland). Variants at 440 Hz were used to build sequences of 6 alternating stimuli consisting of either the syllables "/ba" and "/pi" or the syllables "/co" and "/te." A higher (554 Hz) or lower pitch (440 Hz) syllable was then attached 800 ms following the end of each sequence and represented the target syllable upon which the pitch discrimination was to be judged. Sequences could be either informative, with the first syllable matching the target syllable in identity, or noninformative, that is, the identity of the target syllable could not be anticipated (50% probability) from the first syllable. The time interval between the syllables of a given sequence was also manipulated to be either constant (fixed at 400 ms; rhythmic condition) or pseudorandom (ranging between 100–700 ms; mean 400 ms; nonrhythmic condition). In both cases, the duration of a trial was the same. The overall design, therefore, conformed to a standard $2 \times 2$ orthogonal design with factors identity (informative/noninformative) and rhythm (rhythmic/nonrhythmic).

Participants were instructed to judge whether the target syllable was either higher or lower in pitch compared with the preceding syllables in the sequence, responding as fast as possible while avoiding errors. Participants were also informed about the identity factor, and they were asked to focus on the identity of the first syllable to anticipate target appearance. No reference

to sequence rhythm was made. Each trial began with a fixation cross of variable duration (500–1,000 ms), followed by the syllable sequence and target syllable. A maximum of 2,000 ms was allowed for response after target presentation. An intertrial interval of 1,900 ms followed responses, after which the next trial started.

Participants ran a set of 16 practice trials (with only rhythmic sequences) in which speed and accuracy feedback was given after each trial. This was included in order to ensure proper understanding of the task. A familiarization session was then completed, consisting of 3 blocks of 32 sequences each. The attention task was completed twice, after rTMS lPL and rTMS POz. Each run consisted of 2 blocks of 32 sequences each. Experimental conditions were manipulated within blocks pseudorandomly. Only correct responses were included for the analysis. Outlier trials per condition on the basis of a mean ± 2.5 SD criterion were also removed from further analysis (rTMS POz: 6.7% of trials; rTMS lPL: 6.3% of trials). A repeated-measures ANOVA was used to assess the effects of the factors (within-subject factors intervention [rTMS lPL, rTMS POz], rhythm [rhythmic, nonrhythmic], and identity [informative/noninformative] and between-subject factor intervention order). An ANOVA for each intervention type was performed, with interactions therein explored via paired-samples $t$ tests.

## fMRI acquisition and apparatus

Functional T2*-weighted images were collected using a General Electric MRI 3T scanner (GE Healthcare, Chicago, IL, USA) and a gradient echo-planar imaging sequence to measure BOLD contrast over the whole brain (repetition time [TR] 2,000 ms, echo time [TE] 30 ms; 35 slices acquired in ascending interleaved order; slice thickness: 3 mm with a 0.3 mm gap, $64 \times 64$ matrix, in plane resolution = $3 \times 3$ mm; flip angle, 90˚). Two main fMRI runs with 325 images were acquired for the rule-learning task. Structural images were collected using the MPRAGE-sequence with the following parameters: TR = 7.3 ms, TE/−TI = 2.6, flip angle = 90˚, FOV = $256 \times 256 \times 160$ mm, spatial resolution = 1 mm$^3$/voxel. Auditory stimuli were presented using an amplifier (Sensimetrics, Malden, MA, USA) attached to the earphones, and visual stimuli were presented using MRI-compatible goggles.

## fMRI preprocessing and regression analysis

Data were preprocessed using Statistical Parameter Mapping software (SPM8, Wellcome Trust Centre for Neuroimaging, University College, London, UK, https://www.fil.ion.ucl.ac.uk/spm/). The 2 functional runs were first realigned, and a mean image of all the EPIs was created. This mean image was then coregistered to the T1 image, which was then segmented into gray and white matter by means of the Unified Segmentation algorithm [71]. The resulting normalization parameters were applied to normalize the whole functional set to the MNI space. Finally, functional EPI volumes were resampled into $2 \times 2 \times 2$ mm voxels and spatially smoothed with an 8-mm FWHM kernel.

An event-related design matrix was specified using the canonical hemodynamic response function. Trial onsets were modeled at the moment of the presentation of the first word in a sentence. Two main conditions were modeled: rule and no rule. A temporal first-order modulator and a parametric modulator (we included, for each trial, its RT) were added to each condition. Data were high-pass filtered (to a maximum of 1/90 Hz), and serial autocorrelations were estimated using an autoregressive (AR[1]) model. Remaining motion effects were minimized by also including the estimated movement parameters in the model. First-level contrasts were specified for all participants for the main conditions against the implicit baseline. In addition, a rule versus no-rule contrast was also calculated at the individual level. First-level contrast images were fed to 2 second-level one-sample models with one covariate each to calculate

correlations between brain activity and behavior; rule versus implicit baseline contrasts were correlated with individual learning slopes, and rule versus no-rule contrasts were correlated with the increment in the rule effect in Part 2 over Part 1.

Group results are reported at an FWE $p < 0.05$ corrected threshold at the cluster level with 50 voxels of cluster extent, with an additional uncorrected $p < 0.005$ threshold at the voxel level. Anatomical and cytoarchitectonical areas were identified using the Automated Anatomical Labelling Atlas [72] included in the xjView toolbox (https://www.alivelearn.net/xjview/).

### rTMS protocol

Transcranial magnetic stimulation was delivered through a figure-eight coil attached to a standard MagStim Rapid2 stimulator (maximum stimulation output 2.2 T; Magstim, Whitland, UK). The rTMS was delivered offline, 15 minutes before the rule-learning and the attention tasks, at a frequency of 1 Hz and an intensity of 60% of the maximum stimulation output. This protocol is known to decrease the excitability in the cortical regions beneath the coil position [73] for an average duration of 30 minutes [74] after the end of the stimulation period.

Participants performed 2 rTMS conditions on the same day (Session 2) corresponding to i) the target stimulation site (obtained from the peak activation of the BOLD signal on the lPL; rTMS lPL phase) and ii) the control stimulation site (POz electrode location according to the 10–20 EEG international system; rTMS POz phase). rTMS was applied within an average of 3 mm from that peak.

The control group did not receive rTMS but followed the same behavioral protocol. Because no rTMS was applied, data from L2 and L3 sessions were merged for this group.

When applied onto the lPL, the handle of the coil was angled at 45˚ away from the midline. In contrast, the coil was placed in a vertical position (with the coil handle pointing backwards) for stimulation at POz. Stimulation sites were identified on subject's scalp using the SofTaxic navigator system (EMS, Bologna, Italy) according to the T1 image and a marker of the individual target location, using the coordinates of the functional localizer (as described above). Mean stimulation MNI coordinates located at the lPL were x = −47.85 ± 6.91, y = −39.6 ± 8.62, and z = 48.8 ± 5.52 (see S3 Table for individual coordinates). The SofTaxic navigator system was also used for coil maintenance during the rTMS application (allowed error < 2 mm in 3D space). Because of an experimenter error, 2 subjects were stimulated beyond 3 mm from the peak of activation and were thus excluded from rTMS analyses.

## Supporting information

**S1 Fig. Control group incidental rule-learning task results for Session 1.** Slopes for rule and no-rule blocks over task repetitions derived from the mixed model analysis. The control group showed the expected transition from a significant learning slope in Part 1 ($\beta$diff = −0.8, $t$ = −3.1, $p < 0.002$) to a nonsignificant learning slope ($\beta$diff = 0.06, $t$ = 0.252, $p > 0.8$) with a significant rule effect in Part 2 ($t$[30] = 4.49, $p < 0.001$) and Part 1 ($t$[29] = 3.6, $p < 0.002$). Actual data shown averaged into 6 trial bins (for visual purposes only; the analysis did not bin the data) with the SEM over the slopes for rule and no rule derived from the mixed model analysis. Data used to generate S1 Fig can be found in S4 Data. SEM, standard error of the mean (TIF)

**S2 Fig. Intervention group incidental rule-learning task results for Session 1.** Slopes for rule and no-rule blocks over task repetitions derived from the mixed model analysis. The intervention group—perhaps owing to the smaller sample and/or scanner effects—appeared to comprise slower learners and still showed a significant learning slope during the fMRI phase

(Part 2: βdiff = −0.8, $t$ = −2.63, $p$ < 0.01; Part 1: βdiff = −0.56, $t$ = −2.13, $p$ < 0.034), as well as the expected rule effects (Part 1: $t$[16] = 2.19, $p$ < 0.044; Part 2: $t$[19] = 3.08, $p$ < 0.007). Actual data shown averaged into 6 trial bins (for visual purposes only; the analysis did not bin the data) with the SEM over the slopes for rule and no rule derived from the mixed model analysis. Data used to generate S2 Fig can be found in S5 Data. fMRI, functional MRI; SEM, standard error of the mean
(TIF)

**S3 Fig. Individual fMRI-enhanced activity during later stages of rule learning and rTMS stimulation sites.** In red-yellow, overlap of individual masks for each participant's activation pattern. Only voxels in which at least 10 participants showed individual fMRI-enhanced activity during rule learning are shown. In blue, the sites for rTMS stimulation for all participants is shown (for clarity purposes, 4-mm spheres were created around the stimulation centers). Neurological convention is used, with MNI coordinates shown at the bottom right of each slice. fMRI, functional MRI; MNI, Montreal Neurological Institute; rTMS, repetitive transcranial magnetic stimulation
(TIF)

**S4 Fig. Overlap (yellow) between the ventral network correlating with statistical learning (green) and the main rule effect (rule versus baseline; in green).** In the lPL, there is an overlap between the contrast showing regions in which the BOLD signal significantly covaries with the measure of statistical learning (learning slope) and the contrast showing the brain regions in which activity increases during rule blocks: the more activity in the lPL during rule blocks, the greater (that is, more negative) the slope, and the faster the statistical learning occurs. Only significant results ($p$ < 0.05 FWE-corrected at the cluster level, with an additional $p$ < 0.005 at the voxel level and 50 voxels of cluster extent) are shown over a canonical template with MNI coordinates on the bottom right of each slice. BOLD, blood-oxygenation-level–dependent; FWE, family-wise error; L, Left Hemisphere; lPL, left parietal lobe; MNI, Montreal Neurological Institute
(TIF)

**S1 Table. Whole-brain fMRI activity related to individual differences in rule slopes.** Group-level fMRI local maxima for areas correlating with negative (that is, RTs are still decreasing) rule slopes during the rule blocks in the fMRI phase (see also red-yellow regions in Fig 4A, main text). Results are reported at a FWE $p$ < 0.05 corrected threshold at the cluster level with 50 voxels of cluster extent, with an additional uncorrected $p$ < 0.005 threshold at the voxel level. MNI coordinates were used. BA, Brodmann Area; fMRI, functional MRI; FWE, family-wise error; MNI, Montreal Neurological Institute; RT, reaction time
(DOCX)

**S2 Table. Whole-brain fMRI activity related to individual differences in the rule effect increment for Part 2.** Group-level fMRI local maxima for areas correlating with rule effect increments in Part 2 for the rule versus no-rule contrast during the fMRI phase (see also red-yellow regions in Fig 4B). Results are reported at a FWE $p$ < 0.05 corrected threshold at the cluster level with 50 voxels of cluster extent, with an additional uncorrected $p$ < 0.005 threshold at the voxel level. MNI coordinates were used. BA, Brodmann Area; fMRI, functional MRI; FWE, family-wise error; MNI, Montreal Neurological Institute
(DOCX)

**S3 Table. Coordinates for parietal stimulation sites.**
(DOCX)

**S4 Table. Results for the $d'$ calculated with the order or the dependency violations as false alarms.**
(DOCX)

**S5 Table. Stimuli set for the rule-learning task.**
(DOCX)

**S1 Text. Offline recognition test.**
(DOCX)

**S1 Data. Raw data used to create Fig 3.**
(XLSX)

**S2 Data. Raw data used to create Fig 5.**
(XLSX)

**S3 Data. Raw data used to create Fig 7.**
(XLSX)

**S4 Data. Raw data used to create S1 Fig.**
(XLSX)

**S5 Data. Raw data used to create S2 Fig.**
(XLSX)

## Acknowledgments

We are thankful to A. Correa and D. Sanabria for their helpful insights in the design of the temporal attention task.

## Author Contributions

**Conceptualization:** Joan Orpella, Pablo Ripollés, Manuela Ruzzoli, Julià L. Amengual, Alicia Callejas, Salvador Soto-Faraco, Ruth de Diego-Balaguer.

**Data curation:** Joan Orpella, Pablo Ripollés.

**Formal analysis:** Joan Orpella, Pablo Ripollés, Alicia Callejas.

**Funding acquisition:** Salvador Soto-Faraco, Ruth de Diego-Balaguer.

**Investigation:** Joan Orpella, Pablo Ripollés, Manuela Ruzzoli, Julià L. Amengual, Anna Martinez-Alvarez, Ruth de Diego-Balaguer.

**Methodology:** Joan Orpella, Pablo Ripollés, Manuela Ruzzoli, Julià L. Amengual, Alicia Callejas, Anna Martinez-Alvarez, Ruth de Diego-Balaguer.

**Project administration:** Ruth de Diego-Balaguer.

**Resources:** Manuela Ruzzoli.

**Supervision:** Salvador Soto-Faraco, Ruth de Diego-Balaguer.

**Validation:** Joan Orpella, Pablo Ripollés, Ruth de Diego-Balaguer.

**Visualization:** Joan Orpella, Pablo Ripollés.

**Writing – original draft:** Joan Orpella, Pablo Ripollés, Manuela Ruzzoli, Salvador Soto-Faraco, Ruth de Diego-Balaguer.

**Writing – review & editing:** Joan Orpella, Pablo Ripollés, Manuela Ruzzoli, Julià L. Amengual, Alicia Callejas, Anna Martinez-Alvarez, Salvador Soto-Faraco, Ruth de Diego-Balaguer.

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
