## [Editor Report · Decision Letter 0]

20 Aug 2019

Dear Dr de Diego Balaguer, 

Thank you very much for submitting your manuscript entitled "Integrating When and What Information in the Left Parietal Lobe Allows Language Rule Generalization" for consideration as a Research Article by PLOS Biology. The editors appreciated the attention to an important topic, but we do not feel that the manuscript provides the strength of advance we must require for publication in PLOS Biology. 

We do appreciate the interest to those in the field of your study combining behavioral, functional neuroimaging and transcranial magnetic stimulation to show that language rule learning is a two-stage process involving statistical learning followed by goal-directed attention. Unfortunately, I am sorry to say we are not persuaded that the novel biological insights provided by the study go sufficiently beyond previous work in this area, showing a role of attention for rule-learning and generalization as well as a role of the parietal cortex on attention, to offer the level of significance we aim to publish in PLOS Biology. 

While we cannot consider your manuscript further for publication in PLOS Biology, we suggest, as an alternative, that you consider transferring this manuscript to PLOS ONE (http://journals.plos.org/plosone/). 

PLOS ONE is a peer-reviewed journal that accepts scientifically sound primary research. The review process at PLOS ONE differs from other PLOS journals in that it does not judge the perceived impact of the work or whether this falls within a particular area of research. Rather, it focuses on whether the study has been performed and reported to high scientific and ethical standards, and whether the data support the conclusions. This approach helps to eliminate the rejection cycles that authors commonly encounter when submitting to one journal after another. Please note that the journals are editorially independent and we therefore cannot guarantee the outcome if you choose to pursue a transfer.

If you would like to submit your work to PLOS ONE, please click the following link:

<DeepLinkData><DeepLinkTypeID>27</DeepLinkTypeID><peopleID>608289</peopleID><userSecurityID>fb78f181-2059-4d4d-8b89-2e817a5a0da5</userSecurityID><documentID>37384</documentID><revision>0</revision><manuscriptNumber>PBIOLOGY-D-19-02408</manuscriptNumber><docSecurityID>b61b0502-9c95-4611-99ce-c693f6e43ddd</docSecurityID></DeepLinkData>

If you do NOT wish to submit your work to PLOS ONE, please click this link to decline: 

<DeepLinkData><DeepLinkTypeID>28</DeepLinkTypeID><peopleID>608289</peopleID><userSecurityID>fb78f181-2059-4d4d-8b89-2e817a5a0da5</userSecurityID><documentID>37384</documentID><revision>0</revision><manuscriptNumber>PBIOLOGY-D-19-02408</manuscriptNumber><docSecurityID>b61b0502-9c95-4611-99ce-c693f6e43ddd</docSecurityID></DeepLinkData>

Should you choose to transfer your submission to PLOS ONE, you will receive a confirmation email within 24-48 hours of accepting the transfer. Please note, all PLOS journals are editorially independent and vary in submission requirements. Your submission details and manuscript files will be transferred automatically; once in the PLOS ONE Editorial Manager site, your submission will be returned to you and you will be asked to provide additional information before you can finalize your new submission to PLOS ONE. If you have any questions, please feel free to contact the journal at plosone@plos.org.

Thank you for giving us the opportunity to consider your work.

Sincerely,

Gabriel Gasque, Ph.D.,

Senior Editor

PLOS Biology

---

## [Decision Letter · Decision Letter 1]

29 Oct 2019

Dear Dr de Diego Balaguer,

Thank you very much for submitting your manuscript "Integrating When and What Information in the Left Parietal Lobe Allows Language Rule Generalization" as a Research Article for review by PLOS Biology. As with all papers reviewed by the journal, yours was assessed and discussed by the PLOS Biology editors, by an Academic Editor with relevant expertise, and by four independent reviewers (Reviewer 1, Jutta L. Mueller, has signed her comments). Based on the reviews, I regret that we will not be pursuing this manuscript for publication in the journal. Please accept my sincere apologies for the delay in sending this decision to you.

As you will see from the reviewers' comments, all the referees agree that the multimodal approach used here to assess the acquisition of rule-based language learning is a potentially powerful approach. However, they all raise concerns with core aspects of the task design and analyses that impact the ability to draw strong conclusions about both statistical learning (reviewers 1 and 3) and goal-directed attention (reviewers 2, 3, and 4). Given that these concerns are central to the conclusions of the study, we do not feel that it is appropriate to continue consideration of this study at PLOS Biology.

The reviews are attached, and we hope they may help you should you decide to revise the manuscript for submission elsewhere. I am sorry that we cannot be more positive on this occasion. 

I hope you appreciate the reasons for this decision and will consider PLOS Biology for other submissions in the future. Thank you for your support of PLOS and of Open Access publishing.

Sincerely,

Gabriel Gasque, Ph.D., 

Senior Editor

PLOS Biology

Reviewer remarks:

Reviewer #1, Jutta L. Mueller: The manuscript reports a multimodal study on incidental non-adjacent dependency learning with adult learners undergoing behavioural tasks, fMRI scanning and rTMS stimulation. The learning task comprises the learning of non-adjacent dependencies between two bisyllabic units which are presented in separated three-unit phrases. A word monitoring task was used to measure incidental learning. Functional brain activation was evaluated during learning using different learning-related measures as parametric regressors. The authors identified the ventral and dorsal attention network comprising parietal and frontal regions as related to different aspects of learning. Subsequently rTMS applied to the involved SPL region selectively modulated the application of previously learned rule-knowledge to new material. 

The investigation of on-line rule-learning processes in a multimodal setup is a timely and interesting approach to unravel the brain mechanisms involved in language acquisition. Specifically, the inclusion of rTMS in order to investigate the causal contribution of a specific brain region is great. While I applaud the general approach I had severe upcoming doubts regarding the validity of the results as I was reading through the manuscript in more detail. My two main concerns are i) the framing as a paradigm testing statistical learning and, more seriously, ii) the claim to test the acquisition of non-adjacent dependencies, which is, as far as I can see now, not unambiguously shown by the reported behavioral tests. As these two points concern core aspects of the paper, I recommend substantial revisions before for a potential publication. 

Major

General framing: In their general framing the authors juxtapose statistical learning to attention-based rule learning and claim that their paradigm taps into both processes with statistical learning in initial stages and goal-directed attention in the later stages. While it is difficult to disagree that both processes play a role during the acquisition of complex rules, I have some difficulties applying the concept of statistical learning to the present paradigm. The three-word phrases are presented with jittered pauses during learning as far as I understood the paradigm. This means that there were additional perceptual cues to the to-be-learned structure from the start (namely pauses between the phrases with the crucial elements at the edges). Statistical learning, as it was introduced by Saffran et al., referred to the learning of transitional probabilities between units of speech without any additional acoustic cue. As soon as additional cues are provided learning may not be purely statistical anymore, but guided by other principles of perception. I agree that goal-directed attention must play a larger role as learning proceeds but in my view it is incorrect to subsume the processes before that under the umbrella of statistical learning. 

Another point, which is potentially more related to writing and less to actual content is the argument about the what and when information. How exactly does the current learning task measure the binding of what and when information? In the generalization task the what is new and the when is the same as in the previous learning phases. Looking at that task alone – couldn’t one argue that the when is disturbed by rTMS and for this reason the rule (relating to when) cannot be generalized to the novel items? The parietal lobe might be not necessarily involved in integrating the what and when information in the current study, but rather only in supporting the when information.

Experimental setup: In the learning phases rule-blocks comprise only rule phrases and no-rule blocks comprised different combinations of bisyllabics. Non-adjacent dependency learning is assessed by a word monitoring task where participants are asked to respond to the final word which can be either predicted (in rule blocks) or not predicted (in no-rule blocks). I have a serious concern with regard to the validity of the task. How do we know that it measures non-adjacent dependency learning? From the information in the manuscript it is unclear to me with which frequency the final words occur across the rule and no-rule blocks. As I understand it the final words must be more frequent (independently of the non-adjacent dependency) in the rule blocks compared to the no-rule blocks in which all sorts of elements can occur at the end of a phrase. If the base frequency at the final position for a specific word is increased in the rule blocks it cannot be known anymore whether the RT effect in the word monitoring task is related to mere frequency of that word in the respective position or to the acquired non-adjacent dependency. This means that the rule that was learned could have been much more simple than suggested, namely only that “there is a 50% chance that the target word occurs at the final position” in contrast to “there is a 25% chance that the target word occurs at the final position”. From the manuscript it was unclear to me how exactly the different items are distributed across the blocks. This should be clarified and item lists should be made publicly available (also for replication purposes). If my concern results from a misunderstanding of the design I am happy to accept word monitoring as a measure of incidental non-adjacent dependency learning. But this can hold only when the positional base frequencies of the words were kept constant across blocks.

Relating to this point I am not sure what exactly the offline recognition test measured (and also when exactly it was done – this should be included in the figure (Fig. 1) of the experimental setup).The authors report that three types if test items were used, correct (A1xC1), non-adjacent dependency violations (A1xC2) and swapped (C2xA2). Only the second type of items is useful to assess whether specific non-adjacent dependencies were learned. But the results for this item type are not reported separately and it is not known whether the d’ scores are solely driven by the discrimination of the first and third item type or whether the non-adjacent dependencies were learned as well.

What was the method of correction for multiple comparisons in the fMRI part? How was the minimal cluster size (50 voxels) determined? Please add that information to the manuscript.

Please make the whole list of stimuli available.

Minor

Page 6: mean age = 23.63  23.63 years, please add also for other age specifications.

Page 14: latter  later. 

What is “posterior generalization”?

Page 16: What exactly is a trial? Is one trial a three-word phrase?

Reviewer #2: Orpella and colleagues combined an implicit artificial language learning task with fMRI and inhibitory transcranial magnetic stimulation (TMS) to investigate the role of the left inferior parietal lobe (IPL) in goal-directed attention during language rule generalization. First, healthy subjects underwent a behavioral language learning paradigm insight the MR scanner (or outside for the control group). Thereafter, the fMRI group received inhibitory TMS over either left IPL or a control site and performed learning tasks on two different language sets and completed an attention paradigm. Results were compared within-group and with a control group that did not receive TMS. In the behavioral data, the authors observed a shift from statistical learning to goal-directed attention in later phases. This shift was underpinned by the recruitment of a broad dorsal attention network, including the IPL. TMS over the left IPL impaired the subjects’ ability to generalize learned rules to a structurally analogous new language as well as the ability to integrate stimulus identity and timing in the attention task. Based on these findings, the authors conclude that language rule generalization is a two-step process that engages statistical learning and goal-directed attention, with a key role of the left IPL to integrate stimulus information about identity (what) and timing (when). 

This is a potentially interesting paper that uses a multimodal combination of different approaches (behavioral learning, fMRI, TMS) to tackle the role of the IPL in language rule generalization. While the approach is interesting, I have several major issues that limited my initial enthusiasm about the present study.

1. One major issue with the design was that the control group did not receive any TMS intervention (and no fMRI) and should thus not really be considered as a control group. Why didn’t you include sham stimulation here? Without a sham condition, it is hard to judge whether the observed effects result from a decrease in accuracy after IPL TMS or might also be influenced by a potential facilitation for the active control site. No TMS cannot be regarded as adequate control site because there was simply no intervention, and no side effects. 

2. On a related note, I was surprised that the analyses never directly compared TMS over both sites. I would have expected that the step-down procedure would include a direct comparison of TMS over IPL and the control site (for both the language task and the attention task), but it didn’t. How can you draw strong conclusions if you never directly contrast IPL TMS and control site? Just because IPL TMS disrupts performance and control stimulation doesn’t impair accuracy, it doesn’t mean that the effect for IPL is larger than the effect for the control site. I think this is a main problem with the current study. Direct comparisons are needed here to draw valid conclusions. It also wasn’t clear to me what the interval between the two TMS sessions was – I hope they were not performed on the same day (potential carry-over effects)?

3. TMS results: Couldn't the effects be partially driven by the “no rule” condition, as there is some variation here? The strongest effect appears to be that the control group has much longer RTs. What could be an explanation here? Were the effects corrected for multiple comparisons?

4. Why did you use mixed models for the initital behavioral analyses and ANOVAs for the TMS data? This seems a bit arbitrary.

5. The authors largely ignore the network perspective. While they acknowledge that rule learning and generalization (and attention) engages large-scale fronto-parietal networks, they selectively focus on the left IPL and do not discuss or investigate the interaction between these regions that seem to be crucial for the investigated process of interest. I found it very unfortunate that the fMRI data was simply used to localize the TMS site while the interesting interactions within and between these large-scale networks were ignored.

6. How did you differentiate between learners and non-learners? How was the sample size determined? There is a large difference between the numbers of participants in the different groups – what was the reason for this difference? On a related note, can one really consider the performance insight and outside the scanner as equal (intervention group and control group)? It is stated that two participants from the intervention group were excluded (final sample N=20) because they were stimulated > 2 mm from the appropriate coordinate within the IPL. I don’t understand this, what does that mean? If you have identified the coordinates individually, then why were these inappropriate? With a TMS resolution of approx. 2 cm, I would not consider 2 mm as an exclusion criterion. This all seems a bit arbitrary.

7. The authors largely ignore the large previous literature on artificial grammar learning and the role of the IFG by A. Friederici’s group. While some papers are cited, they are not well integrated or discussed in the discussion section. I think the paper would benefit from including them and discussing similarities and differences between the present study and the previous work. 

8. If you stimulated for 15 minutes, then the after effect should last for approx. 15 minutes after the end of the stimulation, not 30 minutes (see Siebner & Rothwell, 2003). How can you be sure that the effect was still strong enough for both tasks?

Reviewer #3: This paper reports an ambitious and substantial learning study using a combination of behavior, fMRI, and rTMS to try to disambiguate the cognitive/attentional processes underlying the learning of statistical regularities. The line of research is exciting, and potentially of broad interest. 

While there is a lot to recommend the paper, there are some general issues that I think should be addressed. 

-- The theoretical and empirical foundations of the 'Attention' task (using isochronous/anisochronous event timing) are much weaker than the rest of the experiments. The authors make a number of parallels between this task and non-adjacent dependencies in language (mixed rather confusingly across linguistic levels), as well as putative 'what' and 'when' pathways, but there isn't any evidence that these are related. Moreover, this experiment is couched as an 'attention' manipulation but there is no independent evidence that attention is actually being directed implicitly or explicitly, which is a fairly major weakness given that the thrust of the paper is on different aspects of attention. Finally, as the authors note, the results of this experiment are not particularly straightforward. While I understand the general idea behind the experiment (and am sympathetic to it), I would recommend dropping this experiment from the paper as it - at least to my mind - detracts from the main findings, and would require a lot of extra work to properly integrate into the study. 

-- On a related note, in Results, paragraph 2: "The hotspot in IPL was individually identified from the fMRI scans of the secod [typo] part of Session 1, and was assumed to reflect goal-directed attention engaged at a later stage of learning, following Part 1." This assumption needs considerably more grounding. 

-- There is quite a lot of discussion of laterality in the paper (vis-a-vis attentional networks), but no formal tests of laterality, and the rTMS was delivered only to the left. I would recommend pruning these as they do not add to the paper, and indeed confuse the issue as it makes the reader expect that hemispheric differences will be addressed when they are not. 

-- The reaction time measures are (sensibly) the main evidence for online learning. Rather surprisingly given the known problems with this approach, mean RTs are used rather than another measure. I would recommend at least running the statistics with medians or a more sophisticated RT analysis (e.g., diffusion decision or similar). Also, the authors might want to investigate what happens to the regression line when trials are grouped into 4 or 8 bins (for instance). 

-- There is one fairly large anomalous and unaddressed RT finding, namely that the intervention group is so much faster (>100ms) in Session 2 than session 1, while the control group is not. Assuming this is not a mistake in the x-axis of Figure 5, this needs to be tackled. 

-- The structure of the main statistical learning task is extremely simple, and on the face of it, I would have thought that most learning have happened within 15-20 seconds, and would be driven entirely by explicit strategies. Here, "Two different A_C dependencies (A1_C1 and A2_C2) were created out of 4 words from the total word pool." Unless I have misunderstood, in a 'rule' block, the same two word pairs are repeated 48 times (with a single random word intervening between the pair). The words are bisyllabic and with native (Spanish) phonotactics and pronunciation - so learning the two word pairs should almost surely happening within the first 20 or so trials of the first block. 

Despite this, the offline test of this learning (as described in supplementary materials) showed very low average d-prime, which is particularly surprising after the first session where only two word pairs are learned over and over. This might be due to catastrophic interference during the test itself, but did make me wonder whether the participants were learning that the words within each pair were associated, but without attending to order. This would be easy to disambiguate by comparing responses to incorrectly mixed word pairs versus word pairs that were experienced by in the reverse order. 

-- One general issue with the paper is that there is so much data that only some analyses get covered (I am sure to try to avoid reader overload). For instance, in fMRI study, because there are no main effects presented or graphs of the contrasts, it is not possible to know if the correlations are occurring in regions where there is activation, deactivation, or no main effect relative to baseline. For instance, the correlations in the left IPL could be driven by relative decreases in deactivation, as opposed to activation above baseline.

Reviewer #4: In this study rule-learning and generalization was assessed in an artificial language learning task. 22 participants first trained on a first language, while their fMRI activity was recorded. In a second training session the participants learned the same rule in a second and third language, and using rTMS the role of the left parietal cortex during the generalization of this rule was assessed, as well as its role during an additional attentional task. The behavioral results of the intervention group were compared to that of a control group. The authors conclude that for rule-learning there is first a statistical learning stage, followed by a goal-directed attention stage during which the left parietal regions are involved, integrating what & when stimulus information to facilitate rule generalization.

This is a neat study that nicely integrates theoretical views of literature on linguistics and attention. They show that using rTMS stimulation over the parietal cortex interferes with rule generalization, giving novel insight into the neural underpinnings of rule learning. Description of methods and statistics is good.

My main points for improvement concern the interpretation of the results. The authors reason/theorize that during rule-learning there is first a statistical learning stage, followed by a goal-directed attention stage. The interpretation of their results is then consistently in keeping with this terminology. However, the authors have not actually directly assessed goal-directed attention, and I am not convinced that they can label their findings as such. In doing so, they often over-interpret their results. Conclusions need to be tuned down; the results are interesting enough without over-interpretation.

For example: the authors put nice predictions in Figure 2, that then hold up very nicely in the results plotted in Figure 2. On page 6 they then pose: "The combined findings of a non-significant learning slope (significantly flatter than that for Part 1) and a significant rule effect in Part 2 supports the view of a transition from statistical learning to goal-directed attention over successive rule learning parts of Session 1, as predicted.". I would agree that the results nicely show the rule-learning process: first there is statistical learning, and then the rule is known and the rule effect remains. However, in my mind there is no 'goal-directed attention' needed to explain these results. Please be careful in over-interpreting throughout the manuscript, also e.g. on page 13, and at the bottom of page 14: "we have established a causal link between goal-directed attention as engaged in later learning stages and the abstraction and posterior generalization of language rules": this is really too strong: you have established a causal link between left parietal cortex activation in later learning stages and rule generalization: this is really interesting in itself already, but you have not directly tested 'goal-directed attention', so you can not label it as such.

The parietal lobe is chosen as an rTMS stimulation site because of its involvement in temporal goal-directed attention. The resulting effects on behaviour are insightful for its role in rule-learning, but to say you are THUS showing goal-directed attention, is reverse inference. For example, the results shown in Figure 5 in the rTMS LPL group mainly show the rule is not generalized and has to be learned again, but whether this is because of less goal-directed attention, or e.g. because of the rule-being-gone or other transfer/generalization/mapping-difficulties can not really be concluded.

Individual differences in the change in the rule effect from Part 1 to Part 2 were correlated with the rule vs no-rule fMRI contrast, showing a dorsal bilateral fronto-parietal network . This is then interpreted as 'consistent with the engagement of goal directed attention' (page 9). However, as you have correlated an individual difference measure here, doesn't it only mean that for SOME participants (the ones with the largest rule effect increment) there is the involvement of the dorsal bilateral fronto-parietal network? Does this hold at the group-level? Can you also present the results of the main rule effect (rule vs no-rule fMRI contrast) at the group level, to show whether a similar network was active for rule-learning in the whole group? 

With the use of the attention task the authors nicely try to interpret the role of the left parietal lobe during rule learning by looking at its role during another task. As far as I understood from the methods section, the control group also did this attention task. What were their results? Were they comparable to the POz rTMS results, or did the POz rTMS result in again another pattern? Although I do really appreciate the use of the attention task to pin down the exact function of this area, again, be careful with reverse inference: if two tasks are distorted because of rTMS to a particular site, this does not necessarily mean that the two tasks are similar.

---

## [Editor Report · Decision Letter 2]

27 May 2020

Dear Dr de Diego Balaguer,

Thank you for your patience while my colleagues and I reconsidered our decision regarding your manuscript entitled "Integrating When and What Information in the Left Parietal Lobe Allows Language Rule Generalization".

We appreciate the points you raise and are willing to allow you an opportunity to submit your revision. You will appreciate that we cannot make any decision about publication until we have seen the final version of the revised manuscript and your final point-by-point response to the reviewers' comments. Your revision will be sent for external evaluation by our reviewers, and any decision would depend on them being overall supportive and not expressing lingering major technical and/or conceptual concerns. 

***IMPORTANT: We have read your point-by-point response and discussed it with the Academic Editor. We are unsure if it will satisfy reviewer 3's concerns, but have decided that the fairest way forward at this point would be to allow you to submit your revision and ask for reviewer 3's input directly. 

In your revision, you should also cite any additional relevant literature that has been published since the original submission and mention any additional citations in your response. 

We expect to receive your revised manuscript within two months. To submit the revised version, please go to https://www.editorialmanager.com/pbiology/ and log in as an Author. You will find your submission within the folder labelled 'Submissions Needing Revision'. Use the "Submit Revision" link to submit your revised files. 

Before you revise your manuscript, we ask that you please review the following PLOS policy and formatting requirements checklist PDF: https://drive.google.com/file/d/0B_7IflO1bmDTYlZBdmJCT1FUWG8/view?usp=sharing. It's helpful for you to look through the PDF at this stage, and to align your revision with our requirements; should your paper be eventually accepted, this will save everyone time at the acceptance stage.

Now that your manuscript is under active consideration, we ask that you do not submit it elsewhere. Please don't hesitate to contact me if you have any questions or concerns.

Kind regards,

Gabriel Gasque, Ph.D.

Senior Editor

PLOS Biology

---

## [Decision Letter · Decision Letter 3]

12 Aug 2020

Dear Dr de Diego Balaguer,

Thank you for submitting your revised Research Article entitled "Integrating When and What Information in the Left Parietal Lobe Allows Language Rule Generalization" for publication in PLOS Biology. I have now obtained advice from the original reviewers and have discussed their comments with the Academic Editor. You will note that reviewer 1, Jutta Mueller, and 4, Tineke Snijders, have identified themselves. 

Based on the reviews, we will probably accept this manuscript for publication, assuming that you will modify the manuscript to address the remaining points raised by the reviewers. Please also make sure to address the data and other policy-related requests noted at the end of this email.

We expect to receive your revised manuscript within two weeks. 

Your revisions should address the specific points made by each reviewer, but having discussed these comments with the Academic Editor, we will leave it up to you how to best approach these lingering concerns. 

Please submit the following files along with your revised manuscript:

In addition to the remaining revisions and before we will be able to formally accept your manuscript and consider it "in press", we also need to ensure that your article conforms to our guidelines. A member of our team will be in touch shortly with a set of requests. As we can't proceed until these requirements are met, your swift response will help prevent delays to publication.

*Copyediting*

*Published Peer Review History*

*Early Version*

*Submitting Your Revision*

Sincerely,

Gabriel Gasque, Ph.D.,

Senior Editor,

ggasque@plos.org,

PLOS Biology

ETHICS STATEMENT:

-- Please indicate within your manuscript the ID number of the protocol approved by the Universitat de Barcelona, the Universitat Pompeu Fabra (CIEC Parc de Mar) and the European Reseach Council ethics scientific office.

DATA POLICY:

Note that we do not require all raw data. Rather, we ask that for all individual quantitative observations that underlie the data summarized in the figures and results of your paper. For an example see here: http://www.plosbiology.org/article/info%3Adoi%2F10.1371%2Fjournal.pbio.1001908#s5

These data can be made available in one of the following forms:

Regardless of the method selected, please ensure that you provide the individual numerical values that underlie the summary data displayed in the following figure panels: Figures 3, 5, 7, S1, and S2.

Please also ensure that each figure legends in your manuscript include information on where the underlying data can be found and ensure your supplemental data file/s has a legend.

Reviewer remarks:

Reviewer #1, Jutta Mueller: Regarding my concerns with respect to interpretation of the experimentally induced learning processes as statistical learning the authors responded with a rebuttal. Yet, I am not convinced by the response and would like to continue the debate (especially if we would sit together in person). I am a bit drawn between disagreeing with how the term statistical learning is used and the simultaneous impression that this is not a major issue affecting the really good quality and insightfulness of the present study. As the authors cite Jenny Saffran themselves "statistical learning capitalizes on the statistical properties of the language environments". Yet, if pauses structure the input stream it is not only statistical properties of the language environment that is used to capitalize on but also additional perceptual properties. Of course one could argue now that perceptual properties also have a statistical distribution, but then the original idea of statistical learning is somehow distorted, namely that the distributional properties of the information to be learned suffices for learning and not, as I understand it, the distributional properties of the cues to the information to be learned. 

The other problem that I have as a reader relates to the idea of a shift from a learning mechanisms, statistical learning (of which I don't think that it only includes learning of TPs), to goal-directed attention, which can be an ingredient to many different cognitive processes. The two processes, statistical learning and goal-directed attention, are in my view not on the same granularity level. I would conceptualize the present study rather generally as capturing a transition from call-it stimulus-driven learning to attention-driven processing and generalization. Somehow I have the impression that this is not so far from what the authors assume. In the conclusion the authors write about the engagement of stimulus-driven and goal-directed attention - with this it is much easier to agree. Here are several solutions to the problem: 1) The authors could define at the beginning how exactly they use the terms statistical learning and goal-directed attention and/or 2) The authors could just stick with their very consistent interpretation of a two-staged process from learning, which might be more stimulus-driven, to processing and generalization, which might be more driven by endogenous attention. Solution # 3 would be that I accept the authors' interpretation as an instance of a naturally occurring disagreement in science that does not affect the methodological quality of the presented experiment, which would be also fine.

I am fully satisfied with all other responses to my concerns in the last round.

Reviewer #2: The authors have carefully addressed my previous questions and concerns and I think the paper is much stronger now.

Reviewer #3: The authors responded copiously in the rebuttal letter to the reviewer comments, and made a number of changes to the manuscript to address points I had raised in the last review, including the theoretical background for the isochronous/anisochronous sequence task. 

As with R4 in her/his last review, I still do not see a convincing argument that the dependent measure in the 'attention' task necessarily measures what typically is referred to as goal-directed attention. My own view is this would require a fair amount of behavioral work to demonstrate, so not really within the scope of the current study. However, as the authors point out, this type of manipulation has been in the literature on temporal attention. I think readers can make their own decisions on this point. 

Comment and question: Lines 196-198. "The combined findings of a non-significant learning slope (significantly flatter than that for Part 1) and a significant rule effect in Part 2 supports the view of rule-learning as a two stage-process." How is this a 'two-stage process' of learning, as opposed to 'learning' and sustained retention of that learning? Or indeed, just an ongoing process of learning, given this statement: "An analysis of the Intervention and Control groups separately indicated a similar pattern of results in the two groups, except that in the Intervention group some learning was still in progress in Part 2". 

With the new figure showing the fMRI activation related to rule versus non-rule learning (S3 - thanks for creating that), it is unclear how the fMRI results guided the ROI used for the TMS given that the differences in activation were basically found across a broad network of areas. Also, what was the pattern of activation near the coordinates for the POz control site? 

Reviewer #4, Tineke Snijders: The authors addressed my comments satisfactorily. Congratulations on a fine paper!

---

## [Editor Report · Decision Letter 4]

18 Sep 2020

Dear Dr de Diego Balaguer,

On behalf of my colleagues and the Academic Editor, Jennifer K Bizley, I am pleased to inform you that we will be delighted to publish your Research Article in PLOS Biology. 

Early Version

PRESS 

Kind regards,

Vita Usova

Publication Assistant, 

PLOS Biology

on behalf of

Gabriel Gasque,

Senior Editor

PLOS Biology